# Modern temperatures in central–north Greenland warmest in past millennium

M. Hörhold[1✉], T. Münch[2], S. Weißbach[1], S. Kipfstuhl[1], J. Freitag[1], I. Sasgen[1], G. Lohmann[1], B. Vinther[3] & T. Laepple[2,4]

The Greenland Ice Sheet has a central role in the global climate system owing to its size, radiative effects and freshwater storage, and as a potential tipping point[1]. Weather stations show that the coastal regions are warming[2], but the imprint of global warming in the central part of the ice sheet is unclear, owing to missing long-term observations. Current ice-core-based temperature reconstructions[3–5] are ambiguous with respect to isolating global warming signatures from natural variability, because they are too noisy and do not include the most recent decades. By systematically redrilling ice cores, we created a high-quality reconstruction of central and north Greenland temperatures from AD 1000 until 2011. Here we show that the warming in the recent reconstructed decade exceeds the range of the pre-industrial temperature variability in the past millennium with virtual certainty (P < 0.001) and is on average 1.5 ± 0.4 degrees Celsius (1 standard error) warmer than the twentieth century. Our findings suggest that these exceptional temperatures arise from the superposition of natural variability with a long-term warming trend, apparent since AD 1800. The disproportionate warming is accompanied by enhanced Greenland meltwater run-off, implying that anthropogenic influence has also arrived in central and north Greenland, which might further accelerate the overall Greenland mass loss.

Global mean temperature has increased to 1 °C above pre-industrial levels in the second decade of the twenty-first century[6]. Regionally, the Arctic shows the strongest warming[7], particularly in winter[8]. However, to quantify how extraordinary the recently observed temperature changes are, they have to be placed in the context of past temperatures and natural climate variability. For this, instrumental records are often too short, and although climate models are able to reproduce long-term trends[9], they tend to underestimate regional climate variability[10,11] and are challenging to validate. Thus, temperature reconstructions from palaeoclimate proxies are essential for estimating pre-industrial natural climate variability. However, most large-scale reconstructions that are based on multiple proxy types or tree ring records require a proxy screening and instrumental calibration step and thus might be prone to underestimation of past climate variability outside of the calibration period[12].

For the Arctic, the regional temperature reconstruction Arctic 2k[13] shows a persisting warming trend since the nineteenth century and the emergence of air temperature values outside the natural (pre-industrial) variability since the early-mid twentieth century[14]. Increasing temperatures in the Arctic also affect the Greenland Ice Sheet, causing more mass loss by increasing meltwater run-off[1,15]. Weather stations from the Greenland coast cover 200 years[16] and indicate delayed warming trends compared to other regions[17,18] with large regional and seasonal differences[2] along the coast at the beginning of the twenty-first century. Although the melt area of the ice sheet has been observed to progress towards higher elevations[19], little is known about the magnitude and the trend of the surface temperature changes in the central parts of the ice sheet. The reasons lie in the short instrumental records, as well as the sparsity of palaeoclimate data, their low spatial or temporal coverage, and the high noise level in the records.

Previous ice-core data from central and north Greenland provide an inconclusive picture of the imprint of anthropogenic forcing on the surface temperature, either owing to short temporal coverage[3,20], or, because the records are based on single ice-core sites[4], owing to uncertainty on the strength and representability of the contained climate signal[21]. The only available multisite stacked climate record, originating from the North Greenland Traverse (NGT), did not indicate signatures of warming but ended in AD 1995 (ref. [5]).

## The NGT-2012 record

To analyse the Greenland temperature evolution over the past decades with respect to natural variability and global warming, we here extend the previous NGT reconstruction to the year 2011 (all dates are AD). In 2011 and 2012, five of the NGT ice-core sites were revisited and shallow firn cores were taken near the original drilling sites to complement the existing records (Methods). Altogether, the new record ('NGT-2012', Fig. 1a) is stacked from a compilation of 21 stable oxygen isotope records ($\delta^{18}O$ anomalies relative to the 1961–1990 reference interval; Methods) from north and central Greenland (Extended Data Table 1). NGT-2012

[1]Alfred-Wegener-Institut, Helmholtz-Zentrum für Polar- und Meeresforschung, Bremerhaven, Germany. [2]Alfred-Wegener-Institut, Helmholtz-Zentrum für Polar- und Meeresforschung, Potsdam, Germany. [3]Niels Bohr Institute, University of Copenhagen, Copenhagen, Denmark. [4]MARUM – Center for Marine Environmental Sciences and Faculty of Geosciences, University of Bremen, Bremen, Germany. ✉e-mail: maria.hoerhold@awi.de

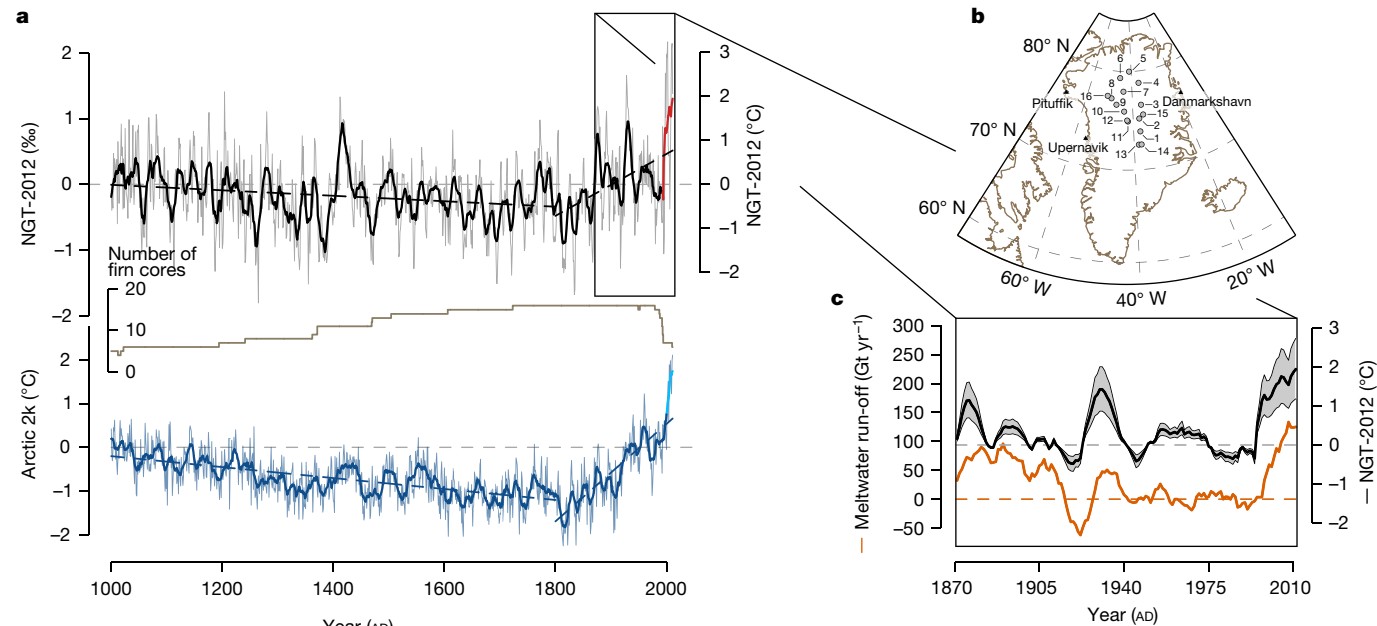

**Fig. 1 | The millennial NGT-2012 temperature reconstruction record from Greenland. a**, The NGT-2012 composite record of 11-year running mean $\delta^{18}O$ (black, left axis) and inferred temperature time series (right axis, Methods) from 1000 to 2011 (top panel). Light grey lines in the background display annual mean values. The thick red line highlights the extension of existing ice-core records to 2011 by re-drillings performed as part of this study. Estimated linear trends over the periods 1000–1800 (pre-industrial) and 1800–2011 are shown as dashed black lines. The number of firn cores contributing to the reconstruction is shown beneath as a brown line. The bottom panel shows the Arctic 2k temperature reconstruction record[13] displayed as 11-year running mean values and as annual data and with dashed blue lines indicating linear trends, as for NGT-2012. The time series was extended to 2011 using HadCrut instrumental data[70] (cyan line, Methods). **b**, Locations of the ice cores used for NGT-2012 (circles) and of nearby weather stations[16] (black triangles; geographic map data obtained from the 'rnaturalearth' package for the software R). Site IDs are detailed in Extended Data Table 1. **c**, Comparison of the NGT-2012 11-year running mean temperature reconstruction (1871–2011, black) with Greenland meltwater run-off from MAR3.5.2[22] ($R = 0.62$, $P < 0.01$, $n = 141$; Methods and Extended Data Fig. 7). Grey shading indicates a ±40% uncertainty of the temperature reconstruction obtained from the range of plausible calibration slopes (Methods). All time series are displayed as anomalies relative to the 1961–1990 reference period (horizontal dashed lines).

covers more than 1,000 years, providing unprecedented spatial and temporal coverage of the area (Fig. 1b). Single records of stable isotope data exhibit a large proportion of non-climatic noise[21]. Here we combine many records to improve the signal-to-noise ratio of our reconstruction (Extended Data Fig. 1). A comparison between individual time series constituting the NGT-2012 stack shows a substantial spatially

coherent signal on decadal and longer timescales with a signal-to-noise ratio greater than 3 (Extended Data Fig. 1). We hence apply an 11-year running mean filter to our time series to focus our analyses on these timescales ('decadal temperatures').

The NGT-2012 stack exhibits a strong correlation ($R \geq 0.75$, $P < 0.01$, $n = 111$) with the decadal annual mean air temperatures from weather

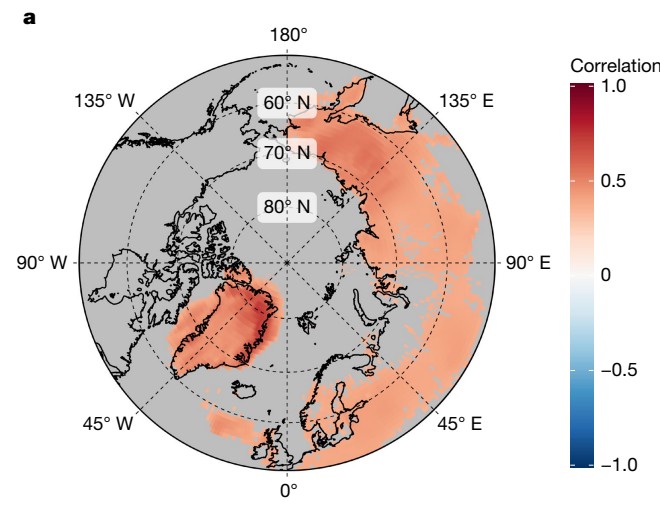

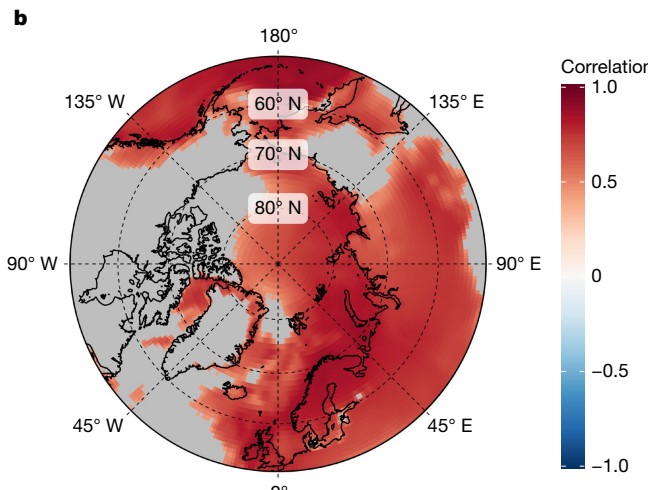

**Fig. 2 | NGT-2012 and Arctic 2k[13] point correlation with the 20CRv3[23,24] near-surface temperature field. a**, Point correlation between the 20CRv3[23,24] reanalysis field of 11-year running mean near-surface temperature and the NGT-2012 11-year running mean $\delta^{18}O$ temperature reconstruction time series. **b**, As in **a** but for the point correlation with the Arctic 2k[13] 11-year running mean temperature reconstruction time series. Correlations are calculated for the time period 1836–2000 for all reanalysis grid cells ≥50° N. Grid cells filled grey mark areas with non-significant correlation values ($P > 0.05$, $n = 165$). All geographic map data are obtained from the 'rnaturalearth' package for the software R.

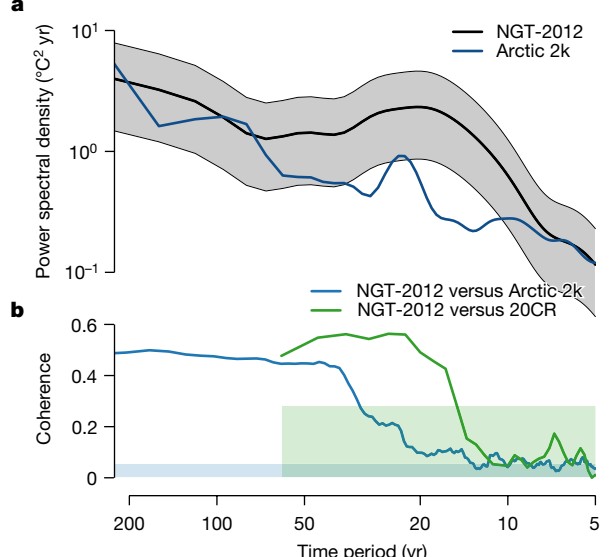

**Fig. 3 | NGT-2012 and Arctic 2k[13] temperature power spectra and coherence.**
**a**, The NGT-2012 spectrum (black) represents the signal content for the
1505–1978 time period, which is common to all individual ice cores from the
stacked record after removing the local noise contribution (Methods); grey
shading denotes the spectral uncertainty range obtained by applying different
plausible temperature calibrations (Methods). The Arctic 2k[13] spectrum (dark
blue) shows the power spectral density of the 1000–2011 time series. Notably,
the average power of the variability in the timescale range from 11 to 51 years is
1.5 to 8 times larger for NGT-2012 than for Arctic 2k (depending on the
temperature calibration). **b**, The magnitude-squared coherence of NGT-2012
and Arctic 2k (blue) and of NGT-2012 and 20CRv3[23,24] averaged across the
region of the NGT-2012 cores (20CR@NGT, green). Blue and green shadings
indicate the respective local 95% confidence level for the coherence based on
surrogate data (Methods).

stations along the Greenlandic coast[16] over the common period (Methods
and Extended Data Fig. 2a). In addition, we find a correlation of $R = 0.76$
($P < 0.001$, $n = 141$) with the decadal annual mean air temperatures at
the coring sites derived from the regional climate model MAR3.5.2[22]
(1871–2011; Extended Data Fig. 2b; Methods). This relationship also
holds true when comparing NGT-2012 to near-surface air tempera-
tures from the Twentieth Century Reanalysis dataset (20CRv3)[23,24]
across the region of the NGT-2012 ice-core locations (20CR@NGT;
$R = 0.62$, $P < 0.005$, $n = 176$; Methods and Fig. 2a). Together, this shows
that the stack can be safely interpreted as a spatially representative
temperature record for central and north Greenland over the past
millennium.

To estimate temperature anomalies, we apply the spatial calibra-
tion slope for Greenland of 1/0.67 °C per ‰ (ref. [25]) and use the range
of published slopes as an uncertainty (Methods and Fig. 1c). This
straightforward approach avoids biases from temporal calibrations
or screening against the instrumental record that affect commonly
used reconstructions[12].

## Natural variability and recent warming

The NGT-2012 temperature record shows a cooling trend from 1000
towards 1800 (−0.06 ± 0.01 °C per 100 years; ±1 standard deviation),
followed by a warming trend until 2011 (0.70 ± 0.11 °C per 100 years;
Fig. 1a). To characterize the natural climate variability on the Greenland
Ice Sheet we analyse the NGT-2012 temperature record using power
spectral analysis including a noise correction for the ice-core stack
(Methods). We find a broad maximum in the spectral power of the
NGT-2012 temperature for time periods from 11 to 51 years, indicating

pronounced natural variability at decadal to bi-decadal timescales
(Fig. 3a).

The reconstructed temperature of the 2001–2011 decade is found
to be on average 1.7 ± 0.4 °C (±1 standard error) warmer than the
1961–1990 reference interval and 1.5 ± 0.4 °C warmer than the twentieth
century (Methods). Despite the pronounced natural variability that we
observe, this high temperature value is exceptional in the context of the
past 1,000 years. The 2001–2011 decadal average of the temperature
anomalies lies clearly outside the distribution of the pre-industrial
values of 1000–1800 (Fig. 4a), with a likelihood for the recent value to
occur under the pre-industrial distribution close to zero ($P = 1.82 \times 10^{-5}$,
Methods). This result is robust against different variations of creating
and analysing the NGT-2012 stack (Extended Data Fig. 3 and Extended
Data Table 2), and holds true for timescales shorter than decadal
(running mean filter windows <11 years; $P \approx 10^{-4}$; Extended Data Table 2).
The recent extreme in temperature can thus be considered as super-
position of the anthropogenic global warming trend and pronounced
natural variability[26,27], which can also explain the ambiguous signatures
of warming, or the lack thereof, in earlier observations from the central
and northern Greenland Ice Sheet[3,5,18,27,28].

## Greenland and Arctic-wide temperatures

The Arctic-wide temperature reconstruction Arctic 2k shows a stronger
cooling trend until 1800 as compared to NGT-2012, and a stronger
warming trend thereafter (Fig. 1a). Throughout the past millennium,
our ice-core-based Greenland temperature reconstruction and the
Arctic-wide temperature reconstruction are correlated ($R = 0.65$,
$P < 0.001$, 1000–2011), but this correlation does not persist when lim-
iting the comparison to the twentieth century ($R = 0.28$, $P = 0.17$, $n = 100$;
Methods and Extended Data Fig. 4), the time period that arguably has
the best reconstruction quality.

To gain further insight into the relationship between NGT-2012 and
Arctic 2k, we analyse their spatial representativeness by calculating
point correlation maps with the 20CRv3 reanalysis temperature data-
set[24] (Fig. 2). This reveals that both reconstructions represent comple-
mentary geographic regions. The Arctic 2k reconstruction represents
large parts of the higher Arctic circumpolar region but only shows a low
correlation over Greenland (Fig. 2b). In a first look, this is surprising
because a number of Greenland ice-core records are included in the
reconstruction. By contrast, the NGT-2012 record exhibits significant
positive regional correlations over the ice sheet (Fig. 2a) and is almost
solely representative for Greenland—a result which is also robust for
annual mean values that are subject to more reconstruction uncertainty
(Extended Data Fig. 5a,b). The distinct spatial correlation structure is
not an artefact of the reconstructions. Replacing the NGT-2012 recon-
struction by the temperature extracted from the 20CRv3 reanalysis
for the region represented by NGT-2012 and Arctic 2k, respectively,
results in virtually the same complementary patterns (Extended Data
Fig. 5c,d). Mechanisms to explain the weak correlation between the
Arctic region and the Greenland Ice Sheet include different elevation[18,28]
and thus different changes in wind, cloud cover or radiation pattern
over the ice sheet[3] and the distinct effect of circulation variability and
changes on Greenland temperature[29,30].

Our results strengthen the observation that the temperature evolu-
tion on the Greenland Ice Sheet is partially separated from that of the
remaining Arctic. This implies that one single time series alone does
not provide a good representation of the Arctic temperature evolution.
Here, our Greenland reconstruction and Arctic 2k together provide a
more complete picture in the assessment of past and recent tempera-
ture changes in the circum-Arctic region and are an important step
towards spatio-temporal reconstructions of the Arctic temperature
evolution.

The decoupling is visible also in the distinct spectrum of tem-
perature variability. Both temperature reconstructions show

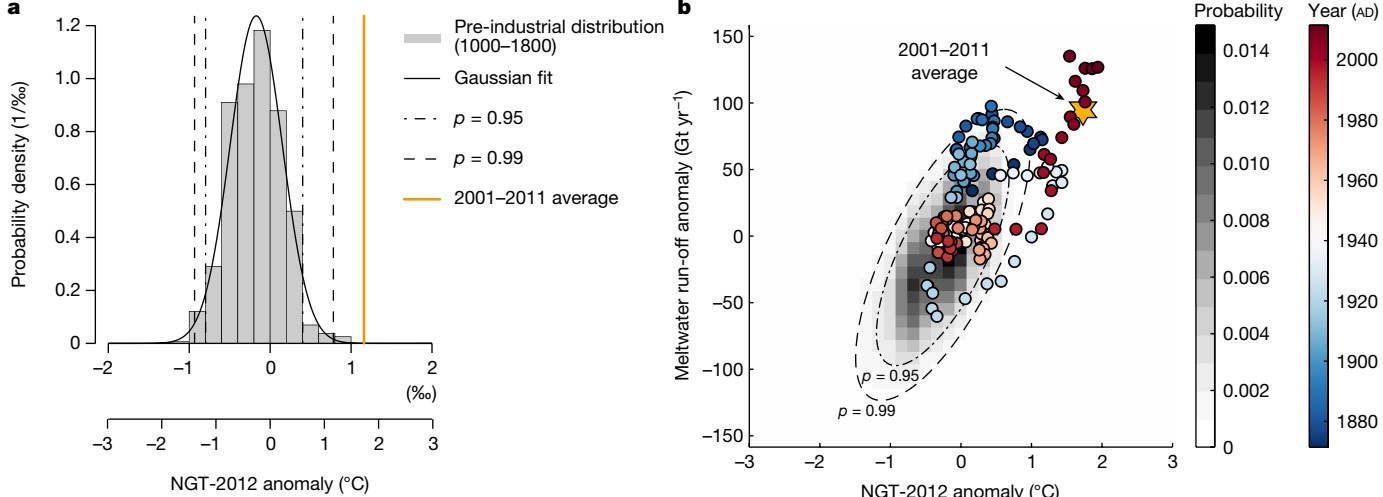

**Fig. 4 | Probability density distributions of past Greenland temperature and meltwater run-off. a**, Histogram of the NGT-2012 11-year running mean $\delta^{18}O$ values and of the related reconstructed temperatures (Methods) for the pre-industrial time period (1000–1800) together with a Gaussian fit (thick black line). Vertical dashed lines show the quantiles corresponding to probabilities of $p = 0.95$ and $p = 0.99$, respectively. The 2001–2011 block-averaged NGT-2012 $\delta^{18}O$ and temperature value is shown as a thick orange line. **b**, Meltwater run-off anomaly from MAR3.5.2[22] as a function of the NGT-2012 temperature anomaly over the common time period 1871–2011 (coloured points;

the 2001–2011 mean value is indicated as the orange star) together with the reconstructed pre-industrial probability density distribution (grey shading). The pre-industrial meltwater run-off distribution is obtained from a linear regression of MAR3.5.2[22] meltwater run-off against NGT-2012 temperature over the common period (Methods; Extended Data Fig. 2g). Dashed ellipses indicate the area corresponding to probabilities of $p = 0.95$ and 0.99, respectively. We note that the high number of outliers is probably due to the increase of NGT-2012 after 1800 (Fig. 1) that might be related to the early onset of industrial-era warming[14].

pronounced decadal to multidecadal variability (11–51 year time periods; Fig. 3a), which is in agreement with other findings for the Arctic region[26,31–33], but for NGT-2012 the variability is four times larger compared to Arctic 2k (range 1.5–8, depending on the temperature calibration). At the same time, both temperature reconstructions exhibit similar power spectral densities for time periods above 50 years and below 8 years. This indicates that the different spatial coverage of the reconstructions is not the primary reason for the variability difference for decadal to multidecadal time periods as a different spatial coverage is expected to mainly affect the short timescales.

Analysing the timescale-dependent relation of the Greenland and Arctic reconstructions shows a high coherence at time periods longer than 50 years, which, however, drops towards shorter time periods (Fig. 3b). By comparison, the coherence between NGT-2012 and the local temperature (NGT@20CRv3) remains high down to time periods below 20 years. This demonstrates that the decoupling between NGT-2012 and Arctic 2k on the decadal to multidecadal timescales is not an artefact of the NGT reconstruction quality.

Thus, the strong temperature variability in the NGT-2012 record probably originates from a regional specific climate signal such as Greenland blocking[29,34,35], an atmospheric variability pattern associated with the negative phase of the North Atlantic Oscillation (NAO). As greater geopotential heights are thermodynamically linked to higher temperatures[30], prolonged atmospheric blocking episodes—that is, persistent high-pressure systems over Greenland—may lead to the northward advection of warm air[36], and accordingly to increased temperatures on the ice sheet[29,36,37]. Indeed, we find a significant correlation ($R = 0.63$, $P < 0.005$, $n = 161$) between the NGT-2012 temperature record and the Greenland Blocking Index (GBI[29]; Extended Data Fig. 2d,e), supporting Greenland blocking as one reason for the larger variability at decadal time scales of the NGT-2012 record compared to Arctic 2k. Greenland blocking was suggested to influence surface melt by influencing the advection of warm air masses[36,37]. In support of this, we find a high correlation between GBI and the Greenland meltwater run-off, derived from the regional climate model MAR3.5.2[22] for the time period 1871–2011 ($R = 0.80$, $P < 0.001$, $n = 141$; Methods and Extended Data Fig. 2f). During

the past decades, the GBI increased in frequency, and to some extent, persistence and magnitude, particularly in summer[37]. This indicates that blocking conditions[34–36], superimposed on thermodynamic warming and natural decadal temperature variability, have contributed to the observed records of summer melt in Greenland.

## Greenland's future meltwater run-off

Greenland has become a major source of mass-related sea level rise[38–41] in the past decade, exceeding thermal expansion and contribution from other glaciers, owing to a strong reduction of its surface mass balance by increased summer melt production[42]. In low-elevation areas, the increased surface air temperatures, changes in albedo and the radiation budget, as well as the decreased capacity of meltwater retention in the firn[1,28,38,43,44], have enhanced meltwater run-off. At the same time, the area undergoing summer melt steadily progresses upwards to higher elevations[19,28,45]. For the period 1871–2011 we find a strong connection ($R = 0.62$, $P < 0.01$, $n = 141$; Methods) between the high-elevation NGT-2012 temperature anomaly and meltwater run-off of the ice sheet. These findings emphasize that increased atmospheric temperatures at high elevations in central and north Greenland are indicative of an increased number and intensity of large melt events, probably also in the future[15]. In principle, the higher meltwater run-off could be partly compensated by an increase of accumulation accompanying the warmer temperatures. Whereas accumulation reconstructions from the NGT-2012 stack are much more uncertain than the NGT-2012-based temperature reconstruction (Methods), they do not provide evidence for a strong link of temperature and accumulation or unprecedented accumulation in the past decade (Extended Data Fig. 6).

The strong statistical and physically meaningful relationship between the NGT-2012 record and the meltwater run-off enables us to generate the first reconstruction of the meltwater run-off anomalies for Greenland over the past millennium and thus to put the recent run-off anomalies into the long-term context (Fig. 4b and Methods). The meltwater run-off anomalies of the 2001–2011 decade are outside the reconstructed distribution of pre-industrial (1000–1800) values taking

into account the reconstruction uncertainties in our linear model. Therefore—although with less certainty than for the temperature—our analysis suggests that current decadal meltwater run-off anomalies are unprecedented over the past millennium. This will probably affect the firn densification and the potential for meltwater storage[19,28,40,46] with further implications for the ice sheet mass balance.

In addition to these findings, our meltwater run-off reconstruction provides a baseline to model past and future freshwater discharge from Greenland[47,48] and their effects on the ocean dynamics, for example the Atlantic Meridional Overturning Circulation[49,50].

Our findings demonstrate that recent temperatures in central and north Greenland are higher than in the past 1,000 years and thus demonstrate that global warming is now also detectable in one of the most remote regions in the world. Likewise, meltwater run-off observed today is probably unprecedented over the past millennium. As warming supports an increased frequency of more widespread summer melt events, reaching in some occasions also central and north Greenland, firn properties such as permeability and meltwater retention may change, comparable to firn changes observed in warmer, and lower-elevation areas. Combined with the finding that temperatures in central and north Greenland and meltwater run-off in the ablation zone are already unprecedented compared to the past millennium, an increasing mass loss of the ice sheet is expected under further global warming.

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

## Methods

### Dataset

We compiled a set of 21 annually resolved records of relative stable isotopic composition ($\delta^{18}O$; that is, the deviation of the ratio of oxygen-18 to oxygen-16 isotopes in the sample from the respective mean ratio in the global ocean, expressed in per mille, and widely used as a temperature proxy) from central and north Greenland (Extended Data Table 1). For all 21 $\delta^{18}O$ records we use the anomaly time series relative to the 1961–1990 mean value in all further analyses. Five of these records are derived from new shallow firn cores obtained between the years 2011 (B26-2012)[51] and 2012 (B18-2012, B21-2012, B23-2012 and NGRIP-2012) to extend the existing $\delta^{18}O$ records originating from the 1993–1995 North Greenland Traverse[5] and from the location of the North Greenland Ice Core Project (NGRIP) deep ice core[52].

The extension cores were measured in the field for di-electrical profiling using the set-up for the North Greenland Eemian Ice Drilling (NEEM) ice core[53] to derive dating tie points by matching against known volcanic eruptions. The cores B18-2012, B21-2012, B23-2012 and NGRIP-2012 were processed and analysed in the cold room facilities of the Alfred Wegener Institute in Bremerhaven, Germany. Firn density was measured by means of two-dimensional X-ray microtomography[54] with a 0.1-mm resolution and the resulting density profiles were smoothed with a Gaussian filter applying a window size of 2 cm. Stable isotopic composition was measured using cavity ring-down spectrometer instruments (L2120-i and L2130-i, Picarro) following the protocol of a previous work[55]. Measurement uncertainty for $\delta^{18}O$ is smaller than 0.1‰. Dating was performed by annual layer counting based on the isotopic composition and the smoothed density profiles, with benchmarking against the identified volcanic events, resulting in an estimated dating uncertainty of ±1 year. The measurement of the isotopic composition and the dating of the extension core B26-2012 was conducted at Copenhagen University. The annual mean $\delta^{18}O$ time series of the extension cores were calculated from the raw $\delta^{18}O$ data over depth and the depth–age relationship, as for the NEGIS core based on the published NEGIS raw data and depth–age relationship. Accumulation rates for the extension cores were derived from the density measurements and the depth–age relationship.

### NGT record extensions

We extend the existing isotope records at the sites B18, B21, B23, B26 and NGRIP, which end in the mid-1990s, with the respective new records until the year 2011. To investigate the reliability of this approach we statistically analysed the overlap period between old and new records considering different running mean filtering window sizes from 1 to 21 years (Extended Data Table 3). The correlation of the annual mean data within the overlap period is somewhat low (≤0.25), probably owing to the strong relative contribution of stratigraphic noise in single records[56], but the correlation systematically increases with increasing window size, with the best correlation observed for 11-year and 21-year filtered data, making the new records faithful representations for the old ones on these timescales.

To account for possible influences from different drilling or measurement techniques, we subtract from the new records the difference in mean isotopic composition within the overlap period (Extended Data Table 3). Starting from the earliest date of the overlap period onwards, the old records are then replaced by the new ones, extending the original records into the year 2011 (2010 for B26), resulting in an effective dataset of 16 $\delta^{18}O$ anomaly records.

### The NGT-2012 isotope stack

We compile our effective dataset of 16 $\delta^{18}O$ anomaly records into a single stack by calculating the simple arithmetic average $\delta^{18}O$ value for each year ('NGT-2012' stack; Fig. 1). Owing to the different lengths of the firn cores and the different accumulation rates at the drill sites,

the total number of firn cores included in the stack changes through time (Fig. 1a). To limit the influence of a very low number of records, we restrict our analyses to the time span 1000–2011, for which the NGT-2012 stack includes a minimum of four records (12 on average).

### Temperature calibration of the NGT-2012 stack

For the conversion from isotopic composition to temperature, linear calibrations exist based either on the relationship of observed present-day spatial gradients in surface snow isotopic composition and temperature (spatial calibration) or on temporal gradients observed at a single site (temporal calibration). Because we work with anomaly time series, we only need to apply a calibration slope (°C per ‰). Here, we use the spatial slope for Greenland of 1/0.67 °C per ‰ (ref. 25) and compare the results to those obtained from using the Holocene temporal slope of 2.1 °C per ‰ from a previous work[57] and the temporal slope for the NEEM site (estimated over 1979–2007) of 1/1.1 °C per mille[20], equivalent to a range of ±40% around the spatial slope. We do not apply any Last Glacial Maximum (LGM)–Holocene temporal slope, as it is not representative for present-day conditions[58] owing to a different seasonality in precipitation or moisture source during the LGM[59,60].

For the spatial slope, we find the last 11 years of the NGT-2012 stack to be on average 1.7 ± 0.4 °C (±1 standard error) warmer than the 1961–1990 reference period and 1.5 ± 0.4 °C warmer than the twentieth century (1901–2000). These values correspond to temperature differences of 2.4 ± 0.6 °C and 2.1 ± 0.6 °C for the temporal slope from ref. 57 and to 1.0 ± 0.2 °C and 0.9 ± 0.2 °C for the NEEM temporal slope[20], showing that the overall uncertainty in the temperature difference when including the uncertainty on the calibration slope is significantly higher than the estimated standard error of the temperature difference itself.

### Firn diffusion

Firn diffusion smooths the isotope signal with increasing strength as a function of time since deposition, described by the diffusion length, until the diffusion process ceases when the firn reaches the density of ice at bubble close-off. As a result, large amplitudes at the surface are damped with increasing depth. We model the diffusion length at each firn core site as a function of depth based on the standard theory for firn diffusion[61], using constant values for the local parameters of annual mean temperature, accumulation rate, surface pressure and surface snow density, as published in the literature[5,52,62–67]. To convert the diffusion lengths from depth into time units, we adopt the Herron–Langway densification model[68].

Owing to the increasing diffusion length, past events of elevated (warm) isotope values might have been stronger initially, that is, prior to diffusion. To assess the impact of firn diffusion on the distribution of the isotopic composition in the NGT-2012 stack, we artificially forward-diffuse each record as if it had been already completely densified to ice by applying a time-dependent differential diffusion length $\sigma(t)$ of

$$\sigma(t) = \sqrt{\sigma_{\text{ice}}^2 - \sigma_{\text{local}}^2(t)},$$

in which $\sigma_{\text{ice}}$ is the modelled diffusion length at the firn–ice transition and $\sigma_{\text{local}}(t)$ is the modelled diffusion length at each time point of the record.

### Spectral analysis

We apply spectral analyses to the isotope records to derive timescale-dependent estimates (power spectral density) of the common climate signal and of the independent local noise, following a previous method[21]. The resulting signal and noise spectra are integrated to compute first the signal-to-noise variance ratio (SNR) as a function of the time resolution of the records and second, based on this, the corresponding expected correlation with the common signal as a function of the number of records averaged[21]. Because the spectral analysis

relies on a fixed number of records for each time point, we restrict the analysis to the time span 1505–1978, which includes 14 of the 16 available records, and which is a trade-off between using many records and covering a sufficiently long time period for the spectral analysis. No diffusion correction is applied to the spectra, but we estimate the timescale range that is most affected by diffusion by determining the critical frequency at which the spectral diffusion transfer function takes a value of $1/e \approx 0.37$. This frequency depends on the value of the diffusion length; adopting the maximum of the estimated diffusion lengths across all isotope records and all observation points in time yields a critical frequency of ~1/7 year$^{-1}$ above which the spectra should be interpreted with care (Extended Data Fig. 1).

We find a distinct local maximum in the variability of the common signal (increased spectral power compared to a power-law background) around the 20-year period (Extended Data Fig. 1a), indicating enhanced climate variability at these timescales. The timescale-dependent estimate of the SNR increases continuously towards longer timescales and scales with the number of records averaged (Extended Data Fig. 1b), ranging from 3.4 at 11-year timescales for the average number of records in the NGT-2012 stack of $n = 12$, compared to 1.1 for $n = 4$ (minimum number) and 4.6 for $n = 16$ (maximum number), to 5.8 at the 100-year period (1.9–7.7). These values correspond to an expected correlation with the common signal at 11-year timescales of 0.73 for averaging $n = 4$ records and $\geq 0.85$ for averaging $n \geq 12$ records (Extended Data Fig. 1b).

We estimate the magnitude-squared coherence between time series to assess their linear relationship as a function of timescale using the smoothed periodogram. Confidence levels are obtained by replacing the original time series with AR1 red-noise surrogate time series with the same autocorrelation and using the frequency averaged $p = 0.95$ sample quantile of $n = 1,000$ realizations.

### Running mean filter and boundary constraints
Prior to the merging of the extended isotope records and the building of the NGT-2012 stack, we apply a running mean filter to each individual record using a window size of 11 years, which is based on the observed correlation within the overlap period of the extended isotope records (Extended Data Table 3), the reasonably high (~0.3) signal-to-noise ratio of a single record at the 11-year timescale (Extended Data Fig. 1b), and avoiding the range of timescales strongly affected by diffusion (Extended Data Fig. 1). To avoid data loss at the time series boundaries from applying the running mean filter, we adopt the 'minimum slope' boundary constraint[69], which is suited for the smoothing of potentially non-stationary time series and which is considered to modestly underestimate the behaviour of the time series near the boundaries in the presence of a long-term trend[69].

### Pre-industrial distribution and comparison to the 2001–2011 time interval
To place the elevated isotope values of the recent 2001–2011 time interval into the historical context of our record, we compute the histogram of the 11-year running mean filtered values of the pre-industrial period (1000–1800). We fit a Gaussian distribution to the histogram, and compare this distribution to the block-averaged value of the recent time interval (Fig. 4a and Extended Data Fig. 3a), finding an extremely low probability for the recent value to occur under the pre-industrial distribution ($P = 1.82 \times 10^{-5}$, Extended Data Table 2).

### The NGT-2012 accumulation rate stack
For an NGT-2012 accumulation rate stack (Extended Data Fig. 6a), we compiled accumulation rate records from the extension cores (B18-2012, B21-2012, B23-2012, B26-2012 and NGRIP-2012) as well as from the cores B16, B18, B21, B26, B29 and NEEM; the data of the remaining cores could not be used owing to insufficient quality. From a spectral analysis equivalent to the one applied to the isotopic data we find a timescale-dependent SNR for the accumulation rate data (Extended

Data Fig. 6b) that is much lower (up to a factor of ~3) than the SNR of the isotopic data. One reason for such a low SNR is the strong spatial variability in local accumulation rates, which affects the accumulation rate reconstructions as local noise, but which can also create long-term artefacts if the spatial variability upstream of the ice-core site affects the down-core record by ice flow. As a result, for NGT-2012 we here use a simple stack of averaging across all available accumulation rate records without first merging the three available pairs of old and extension records, as is done with the isotope data, because the much higher noise level of the accumulation rate data rendered this approach inapplicable. The NGT-2012 isotope and accumulation rate stacks exhibit a low correlation of $R = 0.23$ ($P = 0.05$, $n = 512$) over 1500–2011, as can be expected from the low SNR of the accumulation rate data, without any statistically significant linear relationship (Extended Data Fig. 6a). Even though the NGT-2012 accumulation rate can be seen to have been increasing since 2000, similar to the isotopic data, this time interval is too short to derive any general relationship. In addition, the 2001–2011 block-averaged accumulation rate is not exceptional in the context of the pre-industrial values (Extended Data Fig. 6c), which could be due to noise in the reconstruction or a low sensitivity of the accumulation rate to the recent climate change.

### Comparison with Arctic 2k data
We compare the NGT-2012 isotope stack with the Arctic 2k temperature reconstruction (1–2000)[13]. To cover the full time span of the NGT-2012 stack, we extend the published Arctic 2k record to 2011 with the HadCRUT near-surface instrumental temperature dataset version 5.0.1.0[70] by using the global gridded ensemble mean field of monthly anomalies, computing the annual mean anomalies for each grid cell, taking the area-weighted mean across all grid cells between 60° N and 90° N, and extending the annual Arctic 2k dataset with these data from the year 2001 onwards (Fig. 1a and Extended Data Fig. 4b).

The overall correlation between the extended Arctic 2k reconstruction and the NGT-2012 stack after applying the 11-year running mean filter is $R = 0.65$ ($P < 0.001$, $n = 1,012$; $R = 0.58$, $P < 0.001$, $n = 2,001$ without extension); the correlation over 1901–2011 is $R = 0.66$ ($P < 0.01$, $n = 111$) but only 0.28 ($P = 0.17$, $n = 100$) without extension. A running correlation with a 101-year window size yields a mean correlation of 0.51 and shows variations that overall are within the range expected from surrogate data ($P = 0.84$ that the variations are to be expected by chance), but with unusually low correlation values for the twentieth century (Extended Data Fig. 4c).

The Arctic 2k reconstruction includes the original isotope records from GISP2, GRIP, NGRIP, B16, B18 and B21, which are also used in our compilation. To assess the extent to which these records contribute to the overall Arctic 2k temperature reconstruction, we correlate our extended versions for each of these records with the Arctic 2k record, yielding correlations in the range from 0 to 0.5 (specifically, GRIP: 0.00, GISP2: 0.29, NGRIP: 0.19, B16: 0.39, B18: 0.37 and B21: 0.49; $n = 1,001$). The record from location B21 shows the highest correlation, which is the farthest north and at the lowermost elevation. However, the overall low correlation of these records indicates that their contribution to the Arctic 2k record itself is limited.

### Comparison with instrumental temperature data
We correlate the NGT-2012 isotope stack with nearby instrumental temperature data from the weather stations Upernavik, Pituffik and Danmarkshavn from the Danish Meteorological Institute[16] covering the time period 1873–2011, applying the same 11-year running mean filter to the instrumental temperature data as to the isotope record (Extended Data Fig. 2). We obtain correlation coefficients of $R = 0.87$ (Pituffik, 1948–2011), $R = 0.75$ (Upernavik, 1901–2011) and $R = 0.85$ (Danmarkshavn, 1949–2011) (all $P < 0.005$), which are in the range of expected correlations from our spectral analysis, supporting the interpretation of the isotope stack as a temperature signal for the area. We note that

including the instrumental data from Upernavik prior to 1901 yields a weaker correlation with the NGT-2012 stack, which could be due to limitations of the instrumental data or a weaker representativity of the instrumental record for the area of our firn cores.

### Comparison with reanalysis data

We compute the point correlations of the near-surface temperature field from the Twentieth Century Reanalysis version 3 (20CRv3)[23,24] dataset in the time window 1836–2000 for all grid cells ≥50° N with the NGT-2012 $\delta^{18}O$ anomalies and with the Arctic 2k reconstructed temperature anomalies, using both 11-year running mean as well as annual mean data (Fig. 2 and Extended Data Fig. 5). We specifically rely here on reanalysis data, because no direct instrumental temperature observations exist on the Greenland Ice Sheet and thus observational datasets, such as HadCRUT[71], practically interpolate sea-level-based coastal station data over the ice sheet, leading to spurious correlations. The analyses show that the NGT-2012 record is strongly correlated with the reanalysis temperature over the Greenland Ice Sheet but that the Arctic 2k reconstruction only exhibits nonsignificant correlations there. Although here we focus our analyses on 11-year running mean anomalies, this result is largely robust also for annual mean values.

### MAR3.5.2 surface mass balance and temperature estimates

Greenland meltwater run-off is obtained as a component of the surface mass balance (SMB) output of the regional climate model MAR3.5.2 (Modèle Atmosphérique Régional; version 3.5.2)[22]. Meltwater run-off refers to meltwater production minus meltwater refreezing, deposition and retention. The MAR3.5.2 simulation used here is forced in six-hourly intervals at its lateral boundaries with Twentieth Century Reanalysis version 2 (20CRv2)[23] for the period 1871–2012, and provides 20-km horizontal resolution. This model output is part of a larger number of twentieth-century reconstructions of the Greenland Ice Sheet SMB with MAR3.5.2, forced by various different atmospheric reanalysis datasets[22]. The 20CRv2 forcing is the ensemble mean of a 56-member experimental reanalysis with spatial resolution of 2.0°, assimilating only surface pressure, monthly sea surface temperature and sea ice cover[22].

For the period 1980–2010, MAR3.5.2 forced by 20CRv2 has been shown to exhibit a warm temperature bias (-1 °C) compared to simulations driven by ECMWF Interim reanalysis[72]. However, for the annual meltwater run-off anomalies with respect to 1961–1990 considered in this study, we find that MAR3.5.2/20CRv2 is in good agreement with the latest version MAR3.12 forced by the latest reanalysis (for example, ERA5[73]; Extended Data Fig. 7), within the common period 1950–2012. Even though it is not possible to directly measure mass changes due to meltwater run-off with satellites, we estimate the meltwater run-off anomaly by subtracting net accumulation (snowfall minus sublimation and evaporation) obtained from MAR3.12/ERA5 and ice dynamic discharge obtained from InSAR[42,74] from the GRACE/GRACE-FO annual mass balance with breakpoint January of each year. The results show that the annual variation of the mass budget based on MAR3.12/ERA-5 is consistent with GRACE/GRACE-FO, as is the budget when replacing the meltwater run-off from MAR3.12/ERA5 with the MAR3.5.2/20CRv2 estimates (Extended Data Fig. 7).

For our study, we base the Greenland meltwater run-off anomalies and 2-m surface air temperature data on monthly estimates from MAR3.5.2. The monthly temperature data are sampled at the grid cells closest to the NGT-2012 ice-core locations, averaged across these cells and then averaged to annual mean values; the meltwater run-off data are integrated over the contiguous ice sheet and then cumulated to annual values. Anomalies are calculated with respect to the reference period 1961–1990, which is, first, the commonly used reference period in mass balance studies of the Greenland Ice Sheet[75], and second, synchronous to the one used for the NGT-2012 and Arctic 2k time series. Finally, the same 11-year running mean filter is applied to the annual

temperature values as to the NGT-2012 isotope record, yielding a correlation with the filtered NGT-2012 record over the common time period 1871–2011 of $R = 0.76$ ($P < 0.01$, $n = 141$). Likewise, the correlation of the filtered MAR3.5.2 meltwater run-off anomaly with NGT-2012 is $R = 0.62$ ($P < 0.01$, $n = 141$).

### Comparison with Greenland Blocking Index

We compare the Greenland Blocking Index (GBI)[29] time series to the NGT-2012 temperature and MAR3.5.2 meltwater run-off data over their common time periods. Using 11-year running mean filtered data, the correlation between NGT-2012 and annual GBI is $R = 0.63$ ($P < 0.005$, $n = 161$) and between meltwater run-off and annual GBI it is $R = 0.80$ ($P < 0.001$, $n = 141$). Replacing the annual GBI data with the average GBI for summer (months June, July, August), the correlation with meltwater run-off is $R = 0.91$ ($P < 0.001$, $n = 141$). The correlations are robust also for the unfiltered annual mean values, with correlations of $R = 0.39$ ($P < 0.001$, $n = 161$), $R = 0.56$ ($P < 0.001$, $n = 141$), and $R = 0.67$ ($P < 0.001$, $n = 141$), respectively.

### Significance of correlation between filtered time series

Significance values for the correlation estimates between two running-mean filtered time series (hereafter, 'data' and 'signal') are derived from a Monte Carlo sampling approach, in which $n = 10,000$ realizations ($n = 1,000$ for the correlation maps) of random surrogate data are created with the same AR1 autocorrelation structure as the original (that is, unfiltered) data, filtered with the same running mean filter as the original data, and correlated with the filtered signal. The significance of the observed correlation between filtered data and signal is then obtained from the fraction of surrogate correlations that exceed the observed correlation.

The significance of the running correlation between filtered data and signal is estimated following a method previously described[76]. The correlation between the unfiltered data and signal is used to create $n = 10,000$ random surrogate time series, which exhibit on average the same correlation with the signal as the original data. Surrogate data and the signal are filtered and the running correlation between them is computed. From these surrogate running correlations, we report the local 2.5–97.5% quantiles, and, by expressing the correlation values in terms of $z$ values[76], the overall significance of the variations in the observed running correlation is obtained from the fraction of maximum $z$ value differences for the surrogate data which exceed the maximum $z$ value difference of the observation.

### Sensitivity of probability results

To test the robustness of the found probability for the recent isotope value to occur under the pre-industrial distribution we investigate different variants of creating and analysing the NGT-2012 stack. Specifically, we compare our results based on the main NGT-2012 stack (Fig. 4a) to those obtained for building (1) the NGT-2012 stack from artificially fully forward-diffused data, (2) a stack with a fixed number ($n = 5$) of records through time, (3) a stack from simply averaging across all available isotope records without merging old and new records, (4) as before but including full artificial forward diffusion, and (5) the NGT-2012 stack without adjusting for the difference in mean value within the overlap interval of old and new records (Extended Data Fig. 3 and Extended Data Table 2). All these variants lead to similar probability values for the recent value in the range of $P = 1.8$–$2.6 \times 10^{-5}$ (Extended Data Table 2). For the main NGT-2012 stack, we additionally vary the length of the running mean filter window and the length of the pre-industrial period (shifting it to maximum 1900), which does not affect the probability value notably (all $P \leq 10^{-5}$), except for a running mean filter window of 7, 9 and 21 years ($P \approx 10^{-4}$; Extended Data Table 2). Finally, we adjust the range of the recent period by shifting it into the past in steps of 1 year. This systematically increases the probability value by nearly two orders of magnitude (Extended Data Table 2), which is

expected because the earlier ranges correspond to significantly less elevated isotope values in the NGT-2012 time series (Fig. 1a). We note that the marginal effect of firn diffusion is due to the relatively high accumulation rates at the sites[5] ($\geq$100 kg m$^{-2}$ year$^{-1}$), resulting in small differential diffusion lengths ($\leq$1 year in time units), which have a strong impact on annual and interannual isotope values but only a negligible effect on longer timescales.

### Reconstruction of pre-industrial meltwater run-off distribution

We reconstruct the distribution of meltwater run-off anomalies for the time period of NGT-2012 based on the linear relationship between the NGT-2012 temperatures $T_{core}$ and MAR3.5.2 meltwater run-off $M_{MAR}$ anomalies for the period 1871 and 2012,

$$M_{MAR}^{1871-2012} = T_{core}^{1871-2012}\beta + \epsilon,$$

where $\beta$ is the linear regression coefficient and $\epsilon$ represents uncertainties. We estimate $\hat{\beta}$ and its variance var($\hat{\beta}$) using least-squares adjustment, with the assumption of uniform uncertainties in $M_{MAR}^{1871-2012}$. The reconstructed meltwater run-off $\hat{M}$ for the pre-industrial time period (PI; 1000–1800) based on $T_{core}$ is then obtained as

$$\hat{M} = T_{core}^{PI}\hat{\beta}.$$

To account for uncertainties related to the parameter estimate, as well as the post-fit residual, we calculate the variance of the melt run-off reconstruction as

$$\text{var}(\hat{M}) = \text{var}(M_{MAR}^{1871-2012} - T_{core}^{1871-2012}\hat{\beta}) + \text{var}(\hat{\beta})T_{core}^{PI},$$

using a Monte Carlo approach involving 10,000 random samples.

To derive the two-dimensional distribution of pre-industrial meltwater run-off versus temperature data, we create a 2D grid with 50 bins in each direction spanning the range [$T_1$, $T_2$] and $\hat{\beta}[T_1, T_2]$, where $T_1 = -4$ °C and $T_2 = 4$ °C, and count the number of realizations that fall into each of the bins. The meltwater reconstruction based on the full time period covered by NGT-2012 is obtained by $\hat{M}_{full} = T_{core}\hat{\beta}$.

We note that the finding of the 2001–2011 decade being outside of the pre-industrial distribution is partly a result of this linear reconstruction from the NGT-2012 data, where the 2001–2011 decade is exceptional. The overall run-off is physically not directly linked to temperature, but (1) here we find a linear relationship over the 1871–2012 time period between NGT-2012 and Greenland meltwater run-off and (2) we know that the area affected by melt is changing with changing temperature (increasing under warming conditions). Therefore, we assume that the overall response of the meltwater run-off to changing temperature is linear and thus a linear reconstruction is feasible.

### Data availability

All ice-core stable isotope data used in this study are publicly available in online data repositories from the links given in Extended Data Table 1. The accumulation rate data of the extension cores are available from the respective links for the isotope data; the accumulation rate data of the other used cores can be obtained from the exNGT software project archived under https://doi.org/10.5281/zenodo.7178657. The reconstructed meltwater run-off time series is publicly available at https://doi.org/10.7910/DVN/XQMAY6. Instrumental temperature data for Greenland are available from the Danish Meteorological Institute and were obtained from Polar Portal at http://polarportal.dk/en/news/news/historical-weather-and-climatedata-for-greenland/. Twentieth Century Reanalysis version 3 data are provided by the NOAA/OAR/ESRL PSL, Boulder, CO, USA, at https://www.psl.noaa.gov/data/gridded/data.20thC_ReanV3.html. HadCRUT5 data were obtained from the Met Office Hadley Centre at https://www.metoffice.gov.uk/hadobs/ hadcrut5/index.html. For MARv3.5.2 output we refer to the original publication providing the download link. Source data are provided with this paper.

### Code availability

Software to reproduce the analyses is available as R code hosted in the public Git repository exNGT at https://github.com/EarthSystem-Diagnostics/exNGT, a snapshot of this code (exNGT version 1.0.0) at the time of publication is archived under https://doi.org/10.5281/zenodo.7178657.

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

**Acknowledgements** The re-drilling of the NGT ice cores in 2011 and 2012 was supported by the NEEM ice-core project, which is directed and organized by the Center of Ice and Climate at the Niels Bohr Institute and US NSF, Office of Polar Programs. It is supported by funding agencies and institutions in Belgium (FNRS-CFB and FWO), Canada (NRCan/GSC), China (CAS), Denmark (FIST), France (IPEV, CNRS/INSU, CEA and ANR), Germany (AWI), Iceland (RannIs), Japan (NIPR), South Korea (KOPRI), the Netherlands (NWO/ALW), Sweden (VR), Switzerland (SNF), the UK (NERC) and the USA (US NSF, Office of Polar Programs). We thank the Polar 5 team of the NGT 2012 campaign D. Steinhage, J. Binder, E. Bengtsson, J. Bayes and G. Cirtwill. This project has received funding from the European Research Council (ERC) under the European Union's Horizon 2020 research and innovation programme (grant agreement no.

716092) and was supported by Helmholtz funding through the Polar Regions and Coasts in the Changing Earth System (PACES) programme of the Alfred Wegener Institute as well as through the Helmholtz Research Field Earth and Environment Research Program Changing Earth – Sustaining our Future. We acknowledge support by the Open Access Publication Funds of Alfred-Wegener-Institut Helmholtz-Zentrum für Polar- und Meeresforschung. Support for the Twentieth Century Reanalysis Project version 3 dataset is provided by the US Department of Energy, Office of Science Biological and Environmental Research, by the National Oceanic and Atmospheric Administration (NOAA) Climate Program Office, and by the NOAA Physical Sciences Laboratory. The main plots and numerical analyses in this study were carried out using the open-source software R: A Language and Environment for Statistical Computing. We thank H. Oerter, who supervised the $\delta^{18}O$ measurements of the NGT ice cores. We thank X. Fettweis for publicly providing output of the regional climate model MAR.

**Author contributions** M.H., T.L. and T.M. designed the study and analyses and wrote the initial manuscript draft. T.L. and M.H. oversaw the research. M.H. and T.M. drafted the merging and stacking procedure of the re-drilled ice cores, T.M. developed and wrote the R code. S.W. and J.F. conducted the stable water isotope measurements, J.F. provided the density data and accomplished the dating of the re-drilled ice cores, together with S.W. and T.M. S.K. retrieved the NGT and NGT-2012 cores, B.V. provided all data and dating for B26-2011. I.S. conducted all analysis with MAR and G.L. contributed to the analysis with climate data. All authors contributed to the manuscript.

**Funding** Open access funding provided by Alfred-Wegener-Institut.

**Competing interests** The authors declare no competing interests.

## Additional information
**Correspondence and requests for materials** should be addressed to M. Hörhold.

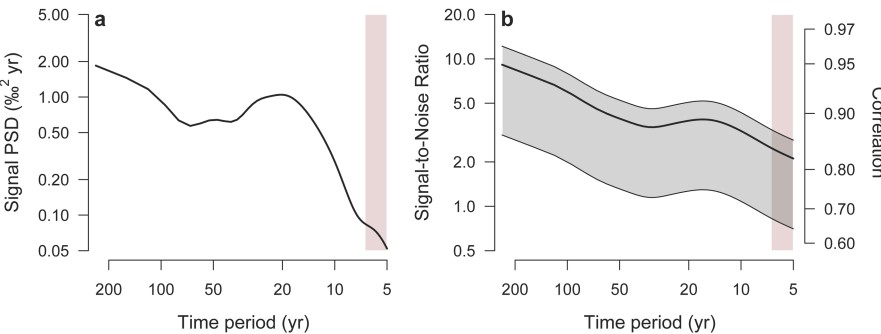

**Extended Data Fig. 1 | Spectral analysis of the NGT-2012 isotope stack.**
**a**, The power spectral density (PSD) of the spatially coherent signal common to all individual firn core records shows enhanced variability around the 20-year period relative to a power-law-type increase towards longer timescales.
**b**, Estimated signal-to-noise ratio of the isotope data as a function of the timescale (left axis) and the corresponding correlation with the common signal (right axis). The thick black line shows the signal-to-noise ratio and the correlation for a stack of $N = 12$ records (the average number of records in the NGT-2012 stack) with the grey-shaded area indicating the range in values for record numbers from $N = 4$ (minimum number in the NGT-2012 stack) to $N = 16$ (maximum number). The red-shaded areas in **a** and **b** indicate the range of timescales strongly influenced by diffusion.

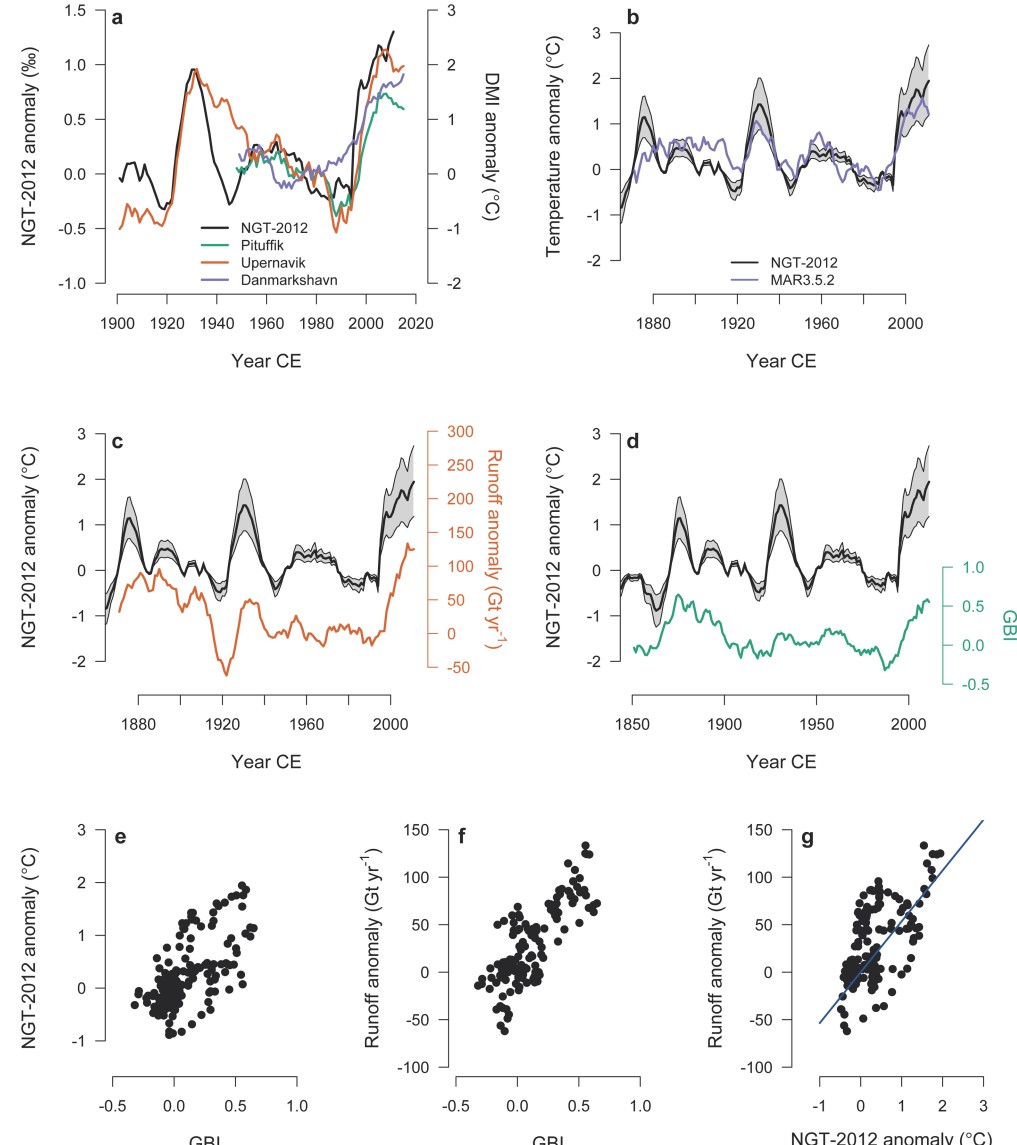

**Extended Data Fig. 2 | Comparison of NGT-2012 with instrumental and model data. a**, NGT-2012 $\delta^{18}$O 11-yr running mean anomalies (black) for the period from 1901 to 2011 together with the 11-yr running mean air temperature anomalies from the three Greenlandic coastal weather stations[16] Pituffik (green), Upernavik (orange) and Danmarkshavn (purple) of the Danish Meteorological Institute (DMI). Correlation coefficients between NGT-2012 and the individual temperature records are R = 0.87 (Pituffik; P < 0.005, n = 64), R = 0.75 (Upernavik; P < 0.005, n = 111) and R = 0.85 (Danmarkshavn; P < 0.005, n = 63). **b**, Comparison of NGT-2012 11-yr running mean temperature anomalies (black), where grey shading denotes a ~40 % uncertainty of the temperature reconstruction obtained from the range of plausible calibration slopes (Methods), with the temperature anomalies from the regional climate model

MAR3.5.2[22] averaged across the NGT-2012 sites (Methods) for the time period 1871–2011. **c**, As **b**, but for the comparison with MAR3.5.2 Greenland meltwater run-off anomalies. **d**, as **b** but for the comparison with the Greenland Blocking Index (GBI)[29]. Correlation coefficients between NGT-2012 temperature and MAR3.5.2 temperature, MAR3.5.2 meltwater run-off and GBI are R = 0.76 (P < 0.001, n = 141), R = 0.62 (P < 0.01, n = 141) and R = 0.63 (P < 0.005, n = 161), respectively. **e**, Scatter plot of NGT-2012 temperature anomalies versus GBI. **f**, Scatter plot of MAR3.5.2 meltwater run-off anomalies versus GBI (correlation R = 0.80, P < 0.001, n = 141). **g**, Scatter plot of MAR3.5.2 meltwater run-off anomalies versus NGT-2012 temperature anomalies with the blue line indicating a least squares linear regression.

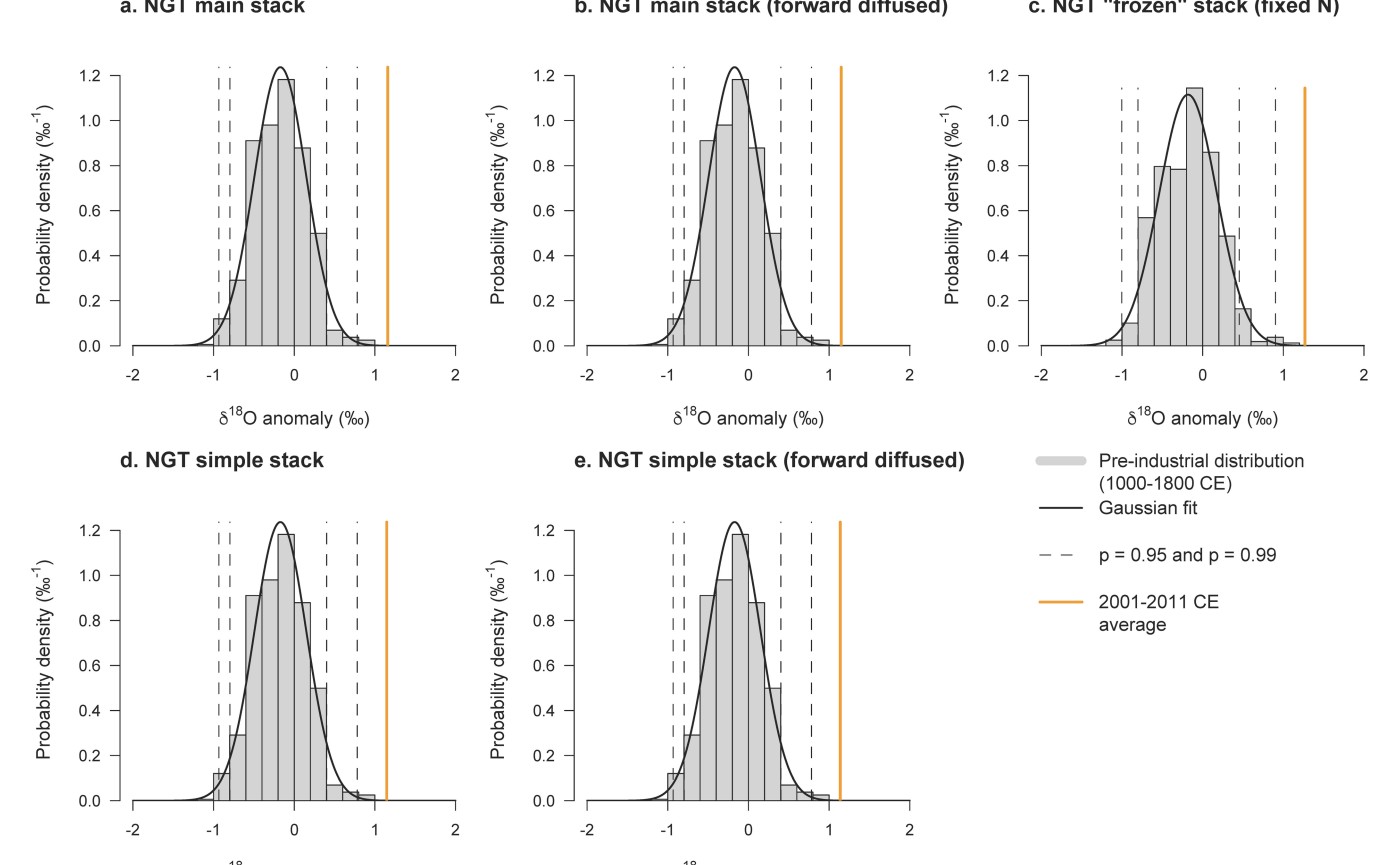

**Extended Data Fig. 3 | NGT-2012 pre-industrial probability density distribution for different analysis variants. a–d,** Compared are our main analysis for the NGT-2012 stack (**a** and Fig. 3) with different analysis variants: the NGT-2012 stack including full forward diffusion of the data (**b**; Methods); a stack built from a fixed number (N = 5) of records through time (**c**); and a simple stack built from averaging across all available isotope records without merging old and new records (**d**). **e,** As in **d** but including full forward diffusion. These variants of deriving and analysing the NGT-2012 stack have an only marginal influence on our results (Extended Data Table 3).

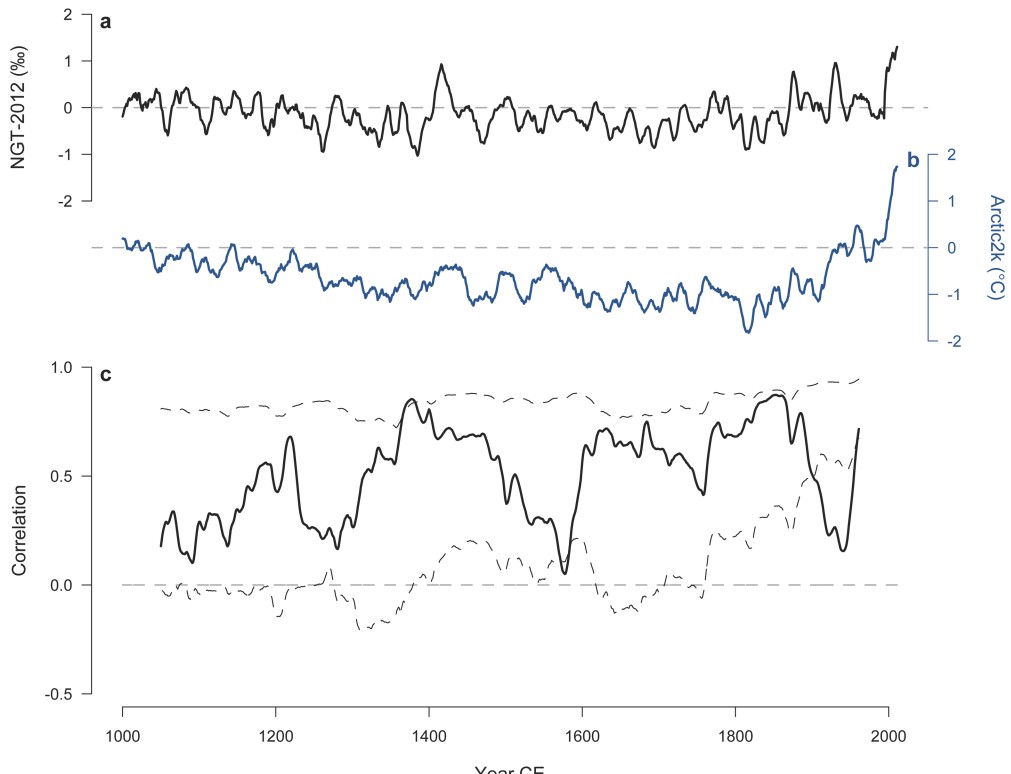

**Extended Data Fig. 4 | Comparison of NGT-2012 with Arctic 2k. a**, NGT-2012 $\delta^{18}O$ 11-yr running mean anomalies for the period from 1000 to 2011 with **b**, 11-yr running mean temperature anomalies relative to the mean in the 1961–1990 reference period from the Arctic 2k temperature reconstruction[13] extended by HadCRUT5[70] instrumental temperature data to the year 2011. **c**, Running correlation over consecutive 101-yr windows between the extended Arctic 2k temperature reconstruction[13] and the NGT-2012 $\delta^{18}O$ data over the time period 1000–2011. The dashed thin black lines show the 2.5–97.5% quantile range of running correlation results for n = 10,000 realizations of surrogate data with the same mean correlation to the Arctic 2k data as the NGT-2012 data, indicating that the observed correlation variations between NGT-2012 and Arctic 2k data are likely to be expected by chance (p = 0.84).

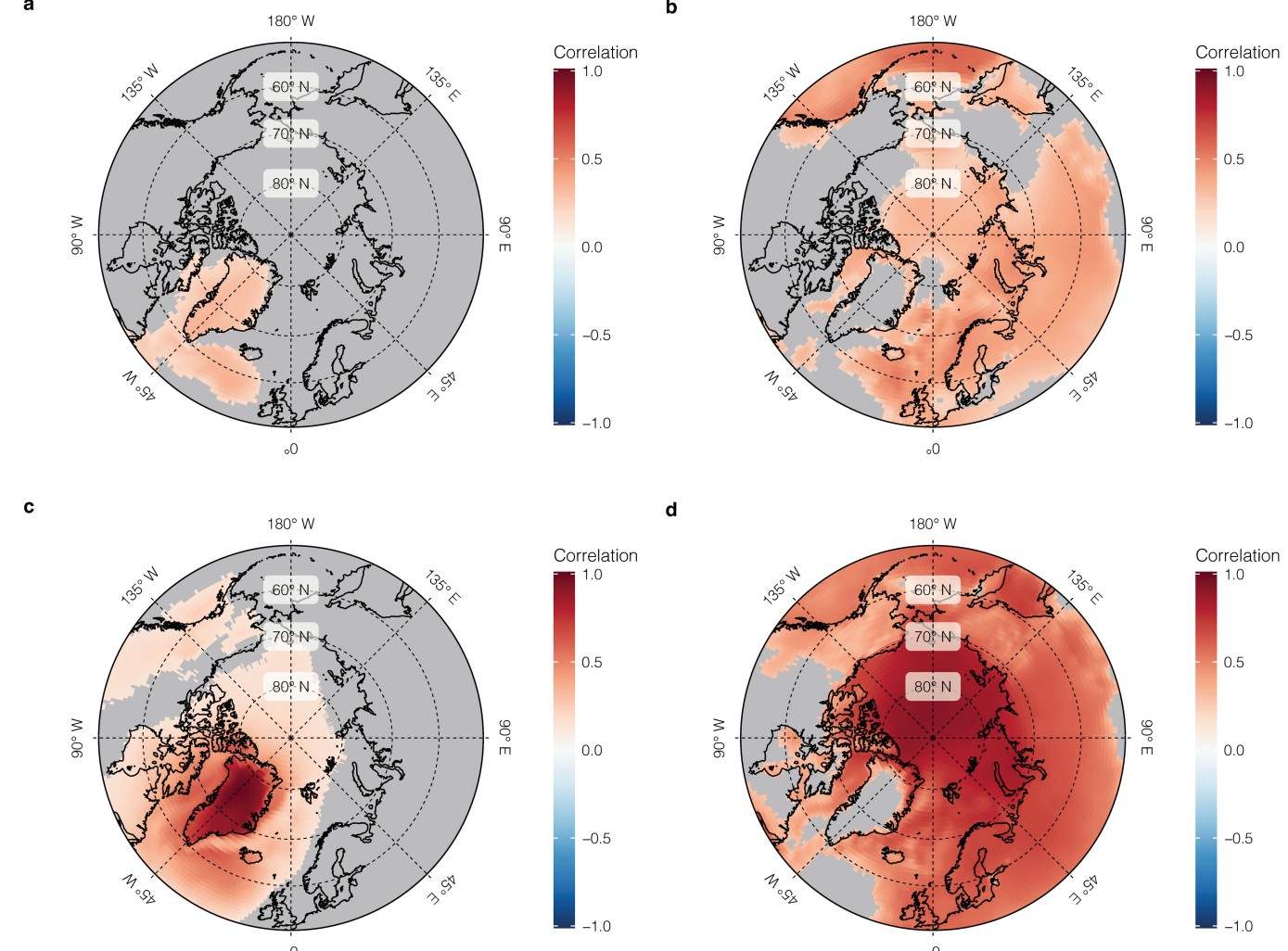

**Extended Data Fig. 5 | NGT-2012 and Arctic 2k point correlation with the Twentieth Century Reanalysis v3 (20CRv3) near-surface temperature field. a**, Point correlation between the 20CRv3 reanalysis field[23,24] of annual mean near-surface temperatures and the NGT-2012 annual mean δ[18]O temperature reconstruction time series. **b**, As in **a** but for the point correlation with the Arctic 2k annual mean temperature reconstruction time series. **c**, As in **a** but for the point correlation with the 20CRv3 reanalysis temperature area mean of the Greenland region. **d**, As in **a** but for the point correlation with the 20CRv3 reanalysis temperature area mean of the Arctic region (60–90° N). Correlations are calculated for the time period 1836–2000 (**a**,**b**) or 1836–2015 (**c**,**d**) for all reanalysis grid cells ≥50° N. Grid cells filled grey mark areas with nonsignificant correlation values (P > 0.05; n = 165 in **a**,**b**; n = 180 in **c**,**d**). All geographic map data are obtained from the 'rnaturalearth' package for the software R.

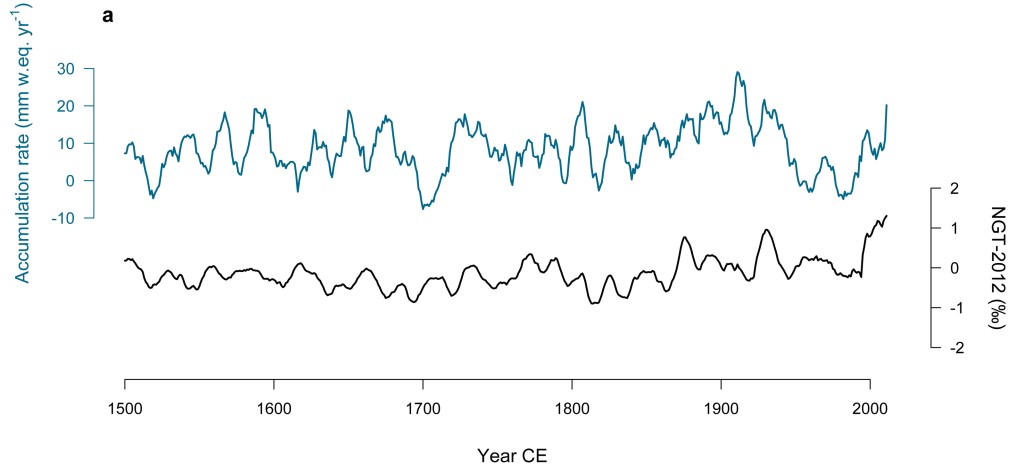

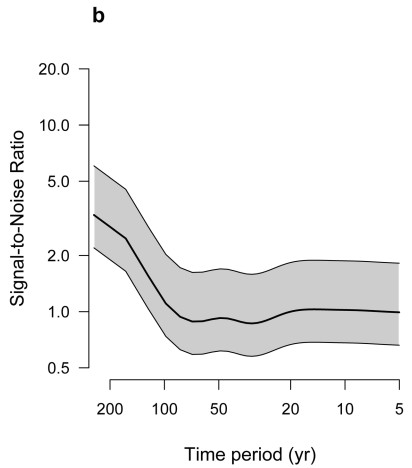

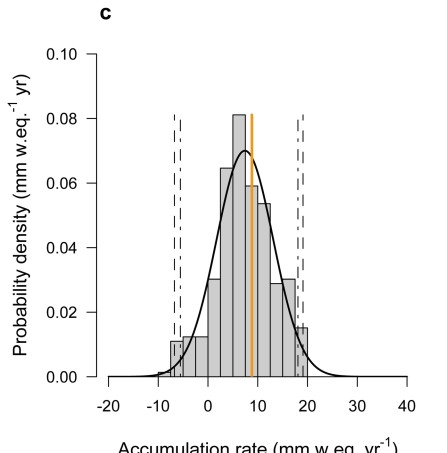

**Extended Data Fig. 6 | NGT-2012 accumulation rate data. a**, NGT-2012 accumulation rate and NGT-2012 δ18O 11-yr running mean anomaly time series from this study for the time period 1500 to 2011. **b**, Estimated signal-to-noise ratio of the NGT-2012 accumulation rate data as a function of the timescale, where the thick black line shows the signal-to-noise ratio for a stack of $N = 6$ records (the average number of records in the NGT-2012 accumulation rate stack) with the grey-shaded area indicating the range in values for record numbers from $N = 4$ (minimum number in the accumulation rate stack) to $N = 11$ (maximum number). **c**, Histogram of the 11-yr running mean NGT-2012 accumulation rate values for the pre-industrial time period (1500–1800) together with a Gaussian fit (thick black line). Vertical black lines show the quantiles corresponding to probabilities of $p = 0.95$ (dash-dotted) and $p = 0.99$ (dashed), respectively. The 2001–2011 block-averaged NGT-2012 accumulation rate value is shown as a thick orange line.

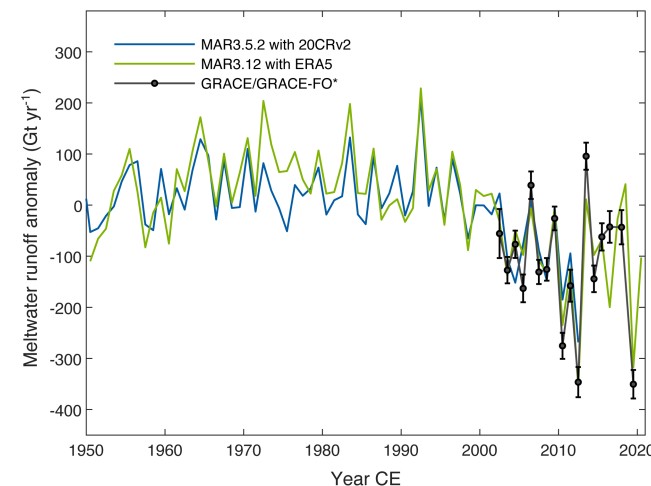

**Extended Data Fig. 7 | Greenland meltwater run-off estimated from model and satellite data.** Shown are the meltwater run-off anomalies for MAR3.5.2[22] forced by 20CRv2[23] (1950–2012; blue), MAR3.12 forced by ERA5[73] (1950–2021; green) and as estimated based on GRACE/GRACE-FO[42,74] data (2002–2020 AD; black). Anomalies for MAR3.5.2 and MAR3.12 are with respect to the years 1961–1990. GRACE/GRACE-FO meltwater run-off anomalies represent the annual mass change measured by GRACE/GRACE-FO minus the net snow accumulation from MAR3.12 and the ice-dynamic discharge from InSAR[74]. An offset was removed to reconcile the mean meltwater run-off anomaly from MAR3.12 and GRACE/GRACE-FO in the period 2002–2020. GRACE/GRACE-FO uncertainties ($2\sigma$) are propagated empirical uncertainties of the mass change only, and do not contain uncertainties associated with the subtracted components net accumulation and ice-dynamic discharge. The reconciliation of the mean mass budget from GRACE/GRACE-FO and details on the GRACE/GRACE-FO data as well as the surface mass balance models are provided in ref. [42].

**Extended Data Table 1 | Overview of firn core records included in the NGT-2012 stack**

| Site ID | Firn core | Time interval (yr CE) | Reference | Data source |
|---|---|---|---|---|
| 3 | B18-2012 | 1865–2011 | this study | https://doi.org/10.1594/PANGAEA.931740 |
| 5 | B21-2012 | 1887–2011 | this study | https://doi.org/10.1594/PANGAEA.931744 |
| 7 | B23-2012 | 1882–2011 | this study | https://doi.org/10.1594/PANGAEA.931745 |
| 12 | NGRIP-2012 | 1965–2011 | this study | https://doi.org/10.1594/PANGAEA.931746 |
| 8 | B26-2012 | 1928–2010 | [51] | https://doi.org/10.1594/PANGAEA.945670 |
| 1 | B16 | 1470–1992 | [5] | https://doi.org/10.1594/PANGAEA.849148 |
| 2 | B17 | 1363–1992 | [5] | https://doi.org/10.1594/PANGAEA.849149 |
| 3 | B18 | 874–1992 | [5] | https://doi.org/10.1594/PANGAEA.849150 |
| 4 | B20 | 775–1993 | [5] | https://doi.org/10.1594/PANGAEA.849152 |
| 5 | B21 | 1372–1993 | [5] | https://doi.org/10.1594/PANGAEA.849153 |
| 6 | B22 | 1372–1993 | [5] | https://doi.org/10.1594/PANGAEA.849154 |
| 7 | B23 | 1023–1993 | [5] | https://doi.org/10.1594/PANGAEA.849155 |
| 8 | B26 | 1505–1994 | [5] | https://doi.org/10.1594/PANGAEA.849156 |
| 9 | B27/28 | 1195–1994 | [5] | https://doi.org/10.1594/PANGAEA.849160 |
| 10 | B29 | 1471–1994 | [5] | https://doi.org/10.1594/PANGAEA.849237 |
| 11 | B30 | 1242–1988 | [5] | https://doi.org/10.1594/PANGAEA.849159 |
| 12 | NGRIP | 0–1995 | [52] | https://www.iceandclimate.nbi.ku.dk/data/Kaufman_etal_2009_data_29sep2009.pdf |
| 13 | GISP2 | 818–1987 | [64] | https://doi.org/10.1594/PANGAEA.55532 |
| 14 | GRIP | 551–1979 | [52] | https://doi.org/10.1594/PANGAEA.786354 |
| 15 | NEGIS | 1607–2011 | [62] | https://doi.org/10.25921/pwgg-j247 |
| 16 | NEEM | 1724–2011 | [20] | https://www.iceandclimate.nbi.ku.dk/data/NEEM_ShallowCore_Pit_Annual_d18O.xlsx |

Listed are the site ID referring to Fig. 1b, the firn core name, the time interval covered by the $\delta^{18}$O annual mean time series, the original reference and the data source. Refs. [5,20,51,52,62,64].

**Extended Data Table 2 | Probability analysis**

| Method | Variant | Pre-industrial period (yr CE) | Recent period (yr CE) | Filter window (yr) | Probability $(10^{-5})$ |
|---|---|---|---|---|---|
| main | | default | default | default | 1.82 |
| main | forward-diffused | default | default | default | 2.08 |
| main | frozen (N = 5) | default | default | default | 2.56 |
| simple | | default | default | default | 2.17 |
| simple | forward-diffused | default | default | default | 2.44 |
| main | non-adjusted mean | default | default | default | 2.17 |
| main | | default | default | 3 | 0.03 |
| main | | default | default | 5 | 0.73 |
| main | | default | default | 7 | 22.4 |
| main | | default | default | 9 | 41.5 |
| main | | default | default | 15 | 0.22 |
| main | | default | default | 21 | 15.1 |
| main | | 1000–1850 | default | default | 1.83 |
| main | merge at start | 1000–1900 | default | default | 2.96 |
| main | merge at end | 1000–1900 | default | default | 2.98 |
| main | | default | 2000–2010 | default | 1.45 |
| main | | default | 1999–2009 | default | 5.60 |
| main | | default | 1998–2008 | default | 8.50 |
| main | | default | 1997–2007 | default | 22.5 |
| main | | default | 1996–2006 | default | 75.1 |

The probability (last column, in $10^{-5}$) for the recent value to occur under the pre-industrial distribution is listed for different parameter combinations of creating and analysing the NGT-2012 stack. Settings named 'default' denote: pre-industrial period = 1000–1800; recent period = 2001–2011; filter window = 11 yr. The first five rows correspond to the different analysis variants of using the stack as in the main text ('main stack'), applying full forward diffusion on the main stack, building the main stack with a fixed number of records through time, and using a simple stack (with and without forward diffusion) built from averaging across all available isotope records without merging old and new records (see also Extended Data Fig. 3).

**Extended Data Table 3 | Overlap statistics**

| Record pair | Overlap interval (yr CE) | Filter window (yr) | SE$_{old}$ (‰) | SE$_{new}$ (‰) | Mean difference (‰) | Correlation |
|---|---|---|---|---|---|---|
| B18 | 1865–1992 | 1 | 0.14 | 0.13 | -0.27 | 0.25 |
| | | 3 | 0.11 | 0.10 | -0.26 | 0.33 |
| | | 5 | 0.10 | 0.09 | -0.26 | 0.38 |
| | | 7 | 0.09 | 0.08 | -0.26 | 0.43 |
| | | 11 | 0.07 | 0.06 | -0.25 | 0.57 |
| | | 21 | 0.06 | 0.04 | -0.23 | 0.75 |
| B21 | 1887–1993 | 1 | 0.14 | 0.15 | 0.04 | 0.16 |
| | | 3 | 0.11 | 0.11 | 0.03 | 0.19 |
| | | 5 | 0.09 | 0.09 | 0.02 | 0.19 |
| | | 7 | 0.08 | 0.07 | 0.02 | 0.19 |
| | | 11 | 0.07 | 0.06 | 0.03 | 0.20 |
| | | 21 | 0.05 | 0.05 | 0.06 | 0.43 |
| B23 | 1882–1993 | 1 | 0.16 | 0.14 | 0.05 | 0.20 |
| | | 3 | 0.13 | 0.10 | -0.03 | 0.17 |
| | | 5 | 0.11 | 0.08 | -0.04 | 0.22 |
| | | 7 | 0.09 | 0.06 | -0.04 | 0.29 |
| | | 11 | 0.07 | 0.05 | -0.04 | 0.41 |
| | | 21 | 0.04 | 0.03 | -0.03 | 0.49 |
| B26 | 1928–1994 | 1 | 0.17 | 0.19 | 0.22 | 0.19 |
| | | 3 | 0.12 | 0.14 | 0.23 | 0.35 |
| | | 5 | 0.09 | 0.12 | 0.23 | 0.43 |
| | | 7 | 0.08 | 0.10 | 0.24 | 0.54 |
| | | 11 | 0.05 | 0.09 | 0.25 | 0.66 |
| | | 21 | 0.03 | 0.07 | 0.31 | 0.71 |
| NGRIP | 1965–1995 | 1 | 0.32 | 0.26 | -0.17 | -0.25 |
| | | 3 | 0.20 | 0.17 | -0.15 | -0.19 |
| | | 5 | 0.15 | 0.13 | -0.13 | -0.15 |
| | | 7 | 0.11 | 0.09 | -0.12 | 0.08 |
| | | 11 | 0.08 | 0.05 | -0.08 | 0.27 |
| | | 21 | 0.06 | 0.04 | -0.04 | 0.24 |
| stack | 1865–1995 | 1 | 0.06 | 0.09 | 0.09 | 0.30 |
| | | 3 | 0.05 | 0.08 | 0.09 | 0.49 |
| | | 5 | 0.04 | 0.07 | 0.09 | 0.59 |
| | | 7 | 0.04 | 0.07 | 0.09 | 0.64 |
| | | 11 | 0.03 | 0.05 | 0.09 | 0.62 |
| | | 21 | 0.02 | 0.03 | 0.11 | 0.47 |

For each pair of old (ending in the early-to-mid 1990s) and new (reaching 2011) records at the same site, the table lists the time interval with overlapping data, the standard error (SE) within the overlap interval for the old and new records, respectively, the mean difference of the records, and their correlation. Standard error, mean difference and correlation are given depending on the length of the running mean filter window used to smooth the data. For 'stack', the record pair consists of stacking all old and all new records separately.