## [Peer review file · Nature]

Manuscript Title: Exceptional temperatures in central-north Greenland relative to the last millenium

Reviewer Comments & Author Rebuttals

Reviewer Reports on the Initial Version:

Referees' comments:

Referee #1 (Remarks to the Author):

See also attached PDF

Exceptional temperatures in central-north Greenland ice cores

Horhold et al.

Horhold et al., present an updated array of north Greenland ice cores which provide valuable spatial and temporal updates to the existing network of ice cores most of which were recovered in the mid 1990s (i.e. the paleoclimate records end at this time). In light of the rapid recent warming observed at Danish Meteorological Institute (DMI) coastal weather stations in Greenland (Extended Data Fig 2), it is critical to extend perspective to high-elevation inland regions of the Greenland ice sheet. The authors demonstrate that the 2001-2011 water stable isotope estimated temperature anomalies in the NGT-2012 compilation are very likely exceptional in the last 1000 years — with important implications for future melting of the Greenland ice sheet.

The authors are careful to evaluate the true signal to noise ratio in the network of ice cores, demonstrating highest SNR at ~11 years, justifying their focus on the data after filtering to that timescale (decadal temperatures).

Such meticulous maintenance/updates to paleoclimate arrays provides the best means for contextualizing recent decades of extreme warmth with past natural variability. Especially in ice core paleoclimate, we are often challenged to evaluate the most recent decades (1990s, 2000s, 2010s) due to a lack of up-to-date cores. This is especially true in Greenland and Antarctica where natural climate variability is fundamentally extreme, challenging such evaluations of recent extremes and trend emergence. Additionally, they clearly demonstrate that the NGT-2012 compilation represent well the conditions across the ice sheet and add credible coverage there, where other climate models lack perspective (e.g. Extended Data Figure 3).

The authors clearly present their methods, from merging new and old ice core records, accounting for diffusion of the isotopic signal, stacking and filtering data, and comparing to climate model / reanalysis output. I find little reason to challenge their approach, but perhaps this is due to my lacking particular skills in such statistical analysis.

This manuscript and the NGT-2012 ice core compilation represent a significant contribution to understanding climate evolution across the Greenland Ice Sheet, especially the emergence of a human-caused warming signal that is of dire consequence for the future mass balance of the ice sheet.

The weakest part of the manuscript is perhaps the conclusion and ties to Greenland meltwater. It is of course obvious that increased air temperatures will lead to increased meltwater and runoff from the ice sheet. While it is useful to point this out, it doesn't seem as though there is much new provided here besides that the ice sheet is warming and will melt more, with mention of how this is likely going to reduce meltwater retention in the firn. The authors may wish to reconsider the goal of this section, and whether it really adds to their excellent presentation of the updated ice core array.

I have relatively few specific in-line comments, but there are some marked in the attached Word document. They may also consider adding a timeline to the title for the timespan of their compilation.

Referee #2 (Remarks to the Author):

Review of Horhold et al: Exceptional temperatures in central-north Greenland ice core

Horhold et al. update a north-central Greenland temperature history derived from a composite of ice-core water isotope records. The decade 2001-2011 is found to be exceptionally warm compared to the previous 1000 years. The north-central Greenland ice core composite (NGT-2012) is compared with the Arctic 2k temperature reconstruction, which has a similar pattern of cooling until 1800 and then warming since, with the warmest period being the 2001-2011 decade. The fundamental conclusion is that the arctic-wide warming also affects the highest elevations of the Greenland ice sheet. The authors suggest that the warm temperature are accompanied by enhanced Greenland meltwater runoff.

The Horhold et al. ice core composite is a welcome update and confirms recent warming is affecting all of the ice sheet. By extending NGT from 1995 to 2010, the recent warming is discernible above the variability. The broader significance of the NGT-2012, however, is not clear. North-central Greenland is a small geographic area which in itself does not play an outsized role in global climate. The freshwater contribution of Greenland is of global importance, but the potential contribution from this region is not discussed and is likely to be a small part of the total. It is already widely accepted that the higher elevations in Greenland have experienced recent warming – e.g. satellite observations of melt area (e.g. Colosio et al., 2021), melt layers (e.g. MacFerrin et al., 2019), and regional climate modeling (e.g. Cullather et al., 2016). Thus, the value of the NGT-2012 seems to be less in the decade-old warming signal of 2001-2010 and more in the ability to place that warming signal in longer context. Unfortunately, the authors do not establish a quantitative relationship between the NGT-2012 mean annual temperature (at decadal timescales) and either Greenland mass balance or atmospheric circulation patterns.

The manuscript has at least three notable gaps.

First, only the annual water isotope records are used. The cores are annually dated and have density profiles, so the annual accumulation rate is also known. Accumulation rates – effectively precipitation in this region of the ice sheet – are a fundamental component of the ice sheet's contribution of sea level. Melt layers have also been used extensively as a relative proxy of summer melt. While I'm not familiar with many of these core sites, NEEM has well documented ice layers and thus many of the cores likely have sufficient melt to have preserved melt layers. Both accumulation rates and melt layers are also useful for interpreting atmospheric circulation patterns; annual accumulation rates because they have shorter correlation length scales than temperature records and melt layers because they are primarily influenced by summer temperature. There is a suggestion that the warm air temperatures are due to atmospheric blocking but the authors provide no new analysis and insight on this potential mechanism for enhanced melt. There is also mention that arctic warming is greatest in the winter and it seems like many of the records in the NGT-2012 stack should have seasonal resolution to address this interesting question.

Second, the relationship between NGT-2012 and melt-water runoff is confusing and not developed statistically. One primary limitation is that the NGT-2012 only appears representative on decadal timescales, preventing annual comparisons. But even with the most recent decade (2001-2010), it is not clear that the MARv3.12 2m air temperature anomalies agree very well with NGT-2012 ones. In Figure 4, the only years I see to the right on the 2001-2010 NGT-2012 average are 2010 and 2003 (just barely). Which means the MARv3.12 decadal average must be considerably cooler. But maybe the larger question is whether annual temperature is even a particularly useful metric for runoff which is largely controlled by summer (melt season) temperature. Thus, it is not clear that the NGT-2012 record is well suited for assessing Greenland melt water runoff.

Third, the NGT-2012 is only compared to Arctic2k, but not any other nearby climate histories. Arctic2k is composed of many records that could be grouped regionally. In particular, there are surrounding ice core records from other locations in Greenland and the Canadian and European Arctic. The nearly opposite spatial correlation maps between NGT-2012 and Arctic2k with 20CR deserves considerably more attention. The greatest concentration of records in Arctic 2k is actually from the same region as NGT-2012. The authors note that these records must not be contributing significantly to Arctic 2k because of the low correlations, but the cause of the lack of influence is not explained. The NGT-2012 and Arctic 2k show the same overall pattern for the last millennium (cooling until ~1800 and then warming), so the question of what timescales the two are related on should be explored more.

The statistics of the NGT-2012 are, overall, well presented. There are some minor issues that could be addressed, like including a subfigure of the number of records at any given time. This information is in the Extended Data Table 1, but is not easy to discern. Where the statistical analysis seems to fall apart is with the relationship to Greenland surface melt – there does not seem to be any statistical analysis of the relationship between NGT-2012 and MARv3.12 runoff. I expected a quantitative relationship to be established between the two such that a record of melt could be extended for the past millennium with NGT-2012. I imagine this was not possible given the lack of annual reliability in the NGT-2012, but without a quantitative relationship, the importance of NGT-2012 on Greenland

melt remains speculative.

Another broad comment is that the figures are not effective. Figure 3 could easily be incorporated into Figure 1, and Figure 2 seems more appropriate for the supplement (or Figure 1 as well). This leaves two full figures that could be dedicated to more detailed analysis.

Specific comments:

L79 – Why are cores from other areas not included if the goal is to “reconstruct and analyze the Greenland temperature evolution over the last decades with respect to natural variability and global warming”. Many other ice core records exist, some of which have been updated like Renland ice cap.

L86 – “unprecedented” – wasn’t there an NGT-1995 which covered this time period already? So unprecedented does not seem accurate.

Figure 3 – This is a clear figure but it would be more effective in Figure 1.

L185 – Arctic 2k uses many of the NGT ice core records. In fact, north-central Greenland has among the greatest density of climate records. How is NGT-2012 filling a spatial gap? Does NGT-2012 changes anything from 1000-1995?

L191 – The stronger temperature variability in NGT-2012 seems like an expected consequence of the small spatial region. What is the expectation for how much greater variability a more geographically limited region should have? Comparing the increased variability to a reasonable expectation would be more informative.

L192 – What is the new evidence for atmospheric blocking in 2001-2010 compared to previous decades with reliable climate reanalyses? It seems like this has been well established by others (e.g. Graeter et al., 2018)

L246 – Can you quantify the impacts on firn densification and meltwater storage? What are the impacts relative to changes in the rest of Greenland?

References:

Colosio, P, M. Tedesco, R. Ranzi, X. Fettweis, 2021. Surface melting over the Greenland ice sheet derived from enhanced resolution passive microwave brightness temperatures (1979-2019). *The Cryosphere*, 15(6), 2623-2646.

Cullather, R.I., S.M.J. Nowicki, B. Zhao, L.S. Koenig, 2016. A characterization of Greenland Ice Sheet surface melt and runoff in contemporary reanalyses and a regional climate model, *Frontiers in Earth Science*, 10.3389/feart.2016.00010

Graeter, K. A., Osterberg, E. C., Ferris, D. G., Hawley, R. L., Marshall, H. P., Lewis, G., et al. (2018). Ice core records of West Greenland melt and climate forcing. *Geophysical Research Letters*, 45, 3164–3172. <https://doi.org>

MacFerrin M., H. Machguth, D. van As, C. Charalampidis, C. M. Stevens, A. Heilig, B. Vandecrux, P. L. Langen, R. Mottram, X. Fettweis, M. R. van den Broeke, W. T. Pfeffer, M. S. Moussavi & W. Abdalati,

2019. Rapid expansion of Greenland's low-permeability ice slabs, *Nature*,
<https://doi.org/10.1038/s41586-019-1550-3>

Author Rebuttals to Initial Comments:

Referee #1 (Remarks to the Author):

See also attached PDF

Exceptional temperatures in central-north Greenland ice cores

Horhold et al.

Horhold et al., present an updated array of north Greenland ice cores which provide valuable spatial and temporal updates to the existing network of ice cores most of which were recovered in the mid 1990s (i.e. the paleoclimate records end at this time). In light of the rapid recent warming observed at Danish Meteorological Institute (DMI) coastal weather stations in Greenland (Extended Data Fig 2), it is critical to extend perspective to high-elevation inland regions of the Greenland ice sheet. The authors demonstrate that the 2001-2011 water stable isotope estimated temperature anomalies in the NGT-2012 compilation are very likely exceptional in the last 1000 years — with important implications for future melting of the Greenland ice sheet.

The authors are careful to evaluate the true signal to noise ratio in the network of ice cores, demonstrating highest SNR at ~11 years, justifying their focus on the data after filtering to that timescale (decadal temperatures).

Such meticulous maintenance/updates to paleoclimate arrays provides the best means for contextualizing recent decades of extreme warmth with past natural variability. Especially in ice core paleoclimate, we are often challenged to evaluate the most recent decades (1990s, 2000s, 2010s) due to a lack of up-to-date cores. This is especially true in Greenland and Antarctica where natural climate variability is fundamentally extreme, challenging such evaluations of recent extremes and trend emergence. Additionally, they clearly demonstrate that the NGT-2012 compilation represent well the conditions across the ice sheet and add credible coverage there, where other climate models lack perspective (e.g. Extended Data Figure 3).

The authors clearly present their methods, from merging new and old ice core records, accounting for diffusion of the isotopic signal, stacking and filtering data, and comparing to climate model / reanalysis output. I find little reason to challenge their approach, but perhaps this is due to my lacking particular skills in such statistical analysis.

This manuscript and the NGT-2012 ice core compilation represent a significant contribution to understanding climate evolution across the Greenland Ice Sheet, especially the emergence of a human-caused warming signal that is of dire consequence for the future mass balance of the ice sheet.

We thank the reviewer for the careful and positive assessment of our study. The revision strongly improved the manuscript and shows the unique information contained in the presented NGT-2012 data set.

Please find the detailed answers and respective changes in the manuscript (colored) to the reviewer comments (black) below. References within our answers are listed at the end. References within the revised text (superscript) refer to references of the manuscript and are not listed here additionally.

The weakest part of the manuscript is perhaps the conclusion and ties to Greenland meltwater. It is of course obvious that increased air temperatures will lead to increased meltwater and runoff from the ice sheet. While it is useful to point this out, it doesn't seem as though there is much new provided here besides that the ice sheet is warming and will melt more, with mention of how this is likely going to reduce meltwater retention in the firn. The authors may wish to reconsider the goal of this section, and whether it really adds to their excellent presentation of the updated ice core array.

Answer:

We agree that the analysis on the link between meltwater run-off and temperature was not presented in-depth. We now provide a detailed statistical analysis of the relationship of

Greenland meltwater runoff and NGT-derived temperatures. The analysis reveals that for the 1871 to 2011 period the NGT-temperature record is strongly related ($R = 0.63$, $p < 0.01$) to the meltwater runoff derived from the regional climate model MAR3.5.2 (Figure 1, Methods) and supported by satellite observations (Extended Data Figure 7). These results lead to new findings:

- The 2001-2011 decade in meltwater runoff, placed in context to instrumental (MAR) and earlier periods, lies outside pre industrial distribution of the last 1000 years (Figure 4)
- The strong statistical and physically plausible relationship allows the use of the NGT-2012 record as a proxy for Greenland meltwater runoff anomalies.

We added the reconstructed meltwater runoff anomalies from the NGT-temperature reconstruction of the pre-industrial period (Figure 4). Such reconstruction will enable model approaches to estimate past and future Greenland meltwater runoff (i.e. see: Lenaerts et al., 2015, Fettweis et al., 2013, Nicholas et al., 2012) and provides an additional constraint on the 1,000 years of surface mass balance relevant for ice-dynamic modeling of the ice sheet. These estimates will also allow study the effect of the freshwater flux into the ocean from the Greenland ice sheet, on the ocean circulation and climate variability (Bamber et al., 2012, Bakker et al., 2016). We therefore decide to keep the section on meltwater runoff in the main part but provide a more in-depth statistical analysis and explanation on the value and impacts of our findings.

Changes in the manuscript:

- We now applied the version 3.5.2 of the MAR model, spanning the time period 1871–2011 CE (previously 1950–2011 CE) to allow a more robust estimate of the statistical relationship.
- We included a subfigure to **Figure 1** showing the common time period 1871–2011 CE of MAR3.5.2 and NGT-2012 to compare the Greenland meltwater run-off and NGT-2012 temperature
- We compared the MAR3.5.2 temperature at the NGT-sites in **Supplement Figure 2**
- We re-structured **Figure 4**, showing the meltwater runoff versus NGT-temperature anomaly for both: the common time period 1871–2011 CE, and based on the linear

coefficient from the regression, the reconstructed meltwater runoff from the NGT-2012 temperature anomalies

- Main, paragraph **Implications for Greenland's future meltwater runoff** we added: *“For the period 1871 - 2011 CE we find a strong connection ($R = 0.63$, $p < 0.01$; Methods) between the high-elevation NGT-2012 temperature anomaly and meltwater runoff of the ice sheet.”*
- Main, paragraph **Implications for Greenland's future meltwater runoff** we added: *“The strong statistical and physically meaningful relationship between the NGT-2012 record and the meltwater runoff allows us to generate the first reconstruction of the meltwater runoff anomalies for Greenland over the last millennium and thus to put the recent runoff anomalies in the long-term context (Figure 4b, Methods). This shows that the meltwater runoff anomalies of the 2001 - 2011 decade are outside the reconstructed distribution of pre-industrial (1000 - 1800 CE) values taking into account the reconstruction uncertainties in our linear model. Therefore, even though with less certainty than for the temperature, our analysis suggests that current decadal meltwater runoff anomalies are unprecedented over the last millennium.”*
- Removed: We compare the NGT-2012 temperature anomalies with Greenland meltwater runoff anomalies from MARv3.12 (Figure 4). We find that the NGT-2012 pre-industrial (1000–1800 CE) temperature anomalies are largely congruent with MARv3.12 for the 1961–1990 CE reference period, commonly associated with a near-balance state (close to zero net mass loss) of the ice sheet⁴⁴. For the most recent decade sampled by NGT-2012 (2001–2011 CE), the temperature anomaly falls within those of the years associated with substantial mass loss in Greenland, including the record meltwater runoff in 2012. While none of the 30 years in the reference period show a meltwater runoff anomaly below -100 Gt yr⁻¹, this threshold is exceeded for 14 out of 21 years after 2000 CE, or eight out of twelve years in the 2000–2011 CE time period of NGT-2012.
- **Methods:** We rephrase the previous section on MAR to: “MAR3.5.2 surface mass balance (SMB): MAR3.5.2 surface mass balance (SMB) and temperature estimates: “Melt water

runoff is obtained as component of the surface-mass balance (SMB) output of the regional climate model MAR3.5.2 (Modèle Atmosphérique Régional; version 3.5.2²¹). Meltwater runoff refers to meltwater production minus meltwater refreezing, deposition and retention. The MAR3.5.2 simulation used here is forced in six-hourly intervals at its lateral boundaries with Twentieth Century Reanalysis version 2 (20CRv2⁶⁸) for the period 1871–2012 CE, and provides 20 km horizontal resolution. This model output is part of a larger number of 20th century reconstructions of the Greenland Ice Sheet SMB with MAR3.5.2, forced by a variety of different atmospheric reanalysis data sets²¹. The 20CRv2 forcing is the ensemble mean of a 56 member experimental reanalysis with spatial resolution of 2.0°, assimilating only surface pressure, monthly sea surface temperature and sea ice cover²¹.

For the period 1980–2010 CE, MAR3.5.2 forced by 20CRv2 has been shown to exhibit a warm temperature bias ($\sim 1^\circ\text{C}$) compared to simulations driven by ECMWF Interim Re-Analysis⁷⁰. However, for the annual melt runoff anomalies with respect to 1961–1990 CE considered in this study, we find that MAR3.5.2 / 20CRv2 is in good agreement with the latest version MAR3.12 forced by the latest reanalysis (e.g., ERA5⁷¹, Extended Data Figure 7), within the common period 1950–2012 CE. Even though it is not possible to measure mass changes due to meltwater runoff directly with satellites, we estimate the meltwater runoff anomaly by subtracting net accumulation (snowfall minus sublimation and evaporation) obtained from MAR3.12 / ERA5 and ice dynamic discharge obtained from InSAR^{41,72} from the GRACE/GRACE-FO annual mass balance with breakpoint January of each year. The results show that the annual variation of the mass budget based on MAR3.12 / ERA-5 is consistent with GRACE/GRACE-FO, as is the budget when replacing the melt runoff from MAR3.12 / ERA5 with the MAR3.5.2 / 20CRv2 estimates (Extended Data Figure 7).

For our study, we base the Greenland meltwater runoff anomalies and 2m surface air temperature data on monthly estimates from MAR3.5.2. The monthly temperature data are sampled at the grid cells closest to the NGT-2012 ice-core locations, averaged across these cells and then averaged to annual mean values; the meltwater runoff data are

integrated over the contiguous ice sheet and then cumulated to annual values. Anomalies are calculated with respect to the reference period 1961–1990 CE which is firstly the commonly used reference period in mass balance studies of the Greenland Ice Sheet⁷³ and secondly synchronous to the one used for the NGT-2012 and Arctic 2k time series. Finally, the same 11-yr running mean filter is applied to the annual temperature values as to the NGT-2012 isotope record over the common time period 1871–2011 CE of $R = 0.77$ ($p < 0.01$). Likewise the correlation of the filtered MAR3.5.2 meltwater runoff anomaly with NGT-2012 is $R = 0.63$ ($p = 0.005$)."

- **Methods:** we add a paragraph on the temperature-meltwater runoff relationship: Reconstruction of pre-industrial of melt runoff distribution: "We reconstruct the pre-industrial (1000–1800 CE) distribution of melt runoff anomalies from the pre-industrial NGT-2012 temperature values based on the observed linear relationship between the NGT-2012 temperatures (T_{core}) and MAR3.5.2 melt runoff (M_{MAR}) anomalies for the period 1871–2012 CE,

$$M_{MAR}^{1871-2012} = T_{core}^{1871-2012} \times \beta + \epsilon,$$

where β is the linear regression coefficient and ϵ represents the residual variability not explained by the linear relationship. We estimate $\hat{\beta}$ and its variance $var(\hat{\beta})$ using least-squares adjustment, with the assumption of uniform uncertainties in $M_{MAR}^{1871-2012}$. The reconstructed melt runoff \hat{M} based on pre-industrial T_{core} is then obtained as

$$\hat{M} = T_{core} \times \hat{\beta}.$$

To allow for uncertainties in the relation, represented by both uncertainties in the parameter estimate as well as in the post-fit residual, we calculate the variance of the melt runoff reconstruction

$$var(\hat{M}) = var(M_{MAR}^{1871-2012} - T_{core}^{1871-2012} \times \hat{\beta}) + var(\hat{\beta}) \times T_{core},$$

using a Monte-Carlo approach involving 10,000 random samples. To derive the two-dimensional distribution of pre-industrial melt runoff versus temperature data, we create a 2D grid with 50 bins in each direction spanning the range $[T_1, T_2]$ and $\hat{\beta} \times [T_1, T_2]$, where $T_1 = -4$ °C and $T_2 = 4$ °C, and count the number of realizations that fall into each of the bins. The finding of the 2001-2011 decade being outside pre-industrial distribution may seem trivial given the linear

dependence on the NGT-2012 data, where the 2001-2011 decade is exceptional. However, as we do consider the uncertainties of the reconstruction, an exceptional NGT-2012 data does not directly imply a statistically significant exceptional runoff. While at a single site, the relation of local temperature and local melt and subsequent runoff is clearly nonlinear, Greenland wide, the warming conditions change the area affected by melt. Thus, it supports our approximation of the linear runoff response across the range of temperatures experienced in the last millennium.”

I have relatively few specific in-line comments, but there are some marked in the attached Word document. They may also consider adding a timeline to the title for the timespan of their compilation.

We thank the reviewer for the helpful comments.

Changes in the manuscript (in response to the marked sections from the reviewer):

- Previous Line 41, now 45: Language: Changed “Reasons are...” to “The reasons lie in”
- Previous Line 44, now 42: Replaced “with” by “at”
- Previous Lines 42, 98, 186: corrected
- Previous Line 193ff: [A3]: Important to highlight dynamically driven warming versus thermodynamically driven warming, both of which may be connected to human-caused warming. Could be more explicit.
 - In Main, paragraph (now renamed to) **Greenland and Arctic wide temperatures:**
We added: *“As greater geopotential heights are thermodynamically linked to higher temperatures²⁹, prolonged (...)”*
We added: *“This indicates that blocking conditions³³⁻³⁵, superimposed on thermodynamic warming and natural decadal temperature variability, have contributed to the observed records of summer melt in Greenland.”*
- Previous Line 226: Is the term “mass-related” commonly used?
Contributors to sea level rise are thermal expansion due to ocean warming and freshwater discharge (Llovel et al., 2019). So the term is used here to refer to the contribution of increasing mass in the ocean (Frederikse et al., 2020), to separate it from

the contribution by thermal expansion. Ocean mass change is related to the ice mass loss from the ice sheets. We thus keep this term here.

- Previous Line 243: insert “reduce”: This sentence was modified to: “*This will likely affect the firn densification and the potential for meltwater storage^{18,26,39,45} with further implications for the ice sheet mass balance.*”
- **Title:** We changed the title to “*Exceptional temperatures in central-north Greenland ice cores relative to the last millennium*”

References (Answer Review 1)

- van den Broeke, M., Bamber, J., Ettema, J., Rignot, E., Schrama, E., van de Berg, W. J., van Meijgaard, E., Isabella Velicognaand, I., Bert Wouters, Science • 13 Nov 2009 • Vol 326, Issue 5955 • pp. 984-986 • DOI: 10.1126/science.1178176
- Trusel, L.D., Das, S.B., Osman, M.B. et al. Nonlinear rise in Greenland runoff in response to post-industrial Arctic warming. Nature **564**, 104–108 (2018). <https://doi.org/10.1038/s41586-018-0752-4>
- Lenaerts, J. T. M., Le Bars, D., van Kampenhout, L., Vizcaino, M., Enderlin, E. M., and van den Broeke, M. R. (2015), Representing Greenland ice sheet freshwater fluxes in climate models, Geophys. Res. Lett., 42, 6373– 6381, doi:10.1002/2015GL064738.
- Fettweis, X., Franco, B., Tedesco, M., van Angelen, J. H., Lenaerts, J. T. M., van den Broeke, M. R., and Gallée, H.: Estimating the Greenland ice sheet surface mass balance contribution to future sea level rise using the regional atmospheric climate model MAR, The Cryosphere, 7, 469–489, <https://doi.org/10.5194/tc-7-469-2013>, 2013
- Nicholas A. Kamenos, Trevor B. Hoey, Peter Nienow, Anthony E. Fallick, Thomas Claverie; Reconstructing Greenland ice sheet runoff using coralline algae. Geology 2012;; 40 (12): 1095–1098. doi: <https://doi.org/10.1130/G33405.1>
- Bamber, J., van den Broeke, M., Ettema, J., Lenaerts, J., and Rignot, E. (2012), Recent large increases in freshwater fluxes from Greenland into the North Atlantic, Geophys. Res. Lett., 39, L19501, doi:10.1029/2012GL052552.

- Bakker, P., et al. (2016), Fate of the Atlantic Meridional Overturning Circulation: Strong decline under continued warming and Greenland melting, *Geophys. Res. Lett.*, 43, 12,252– 12,260, doi:10.1002/2016GL070457.)
- (Frederikse, T., Landerer, F., Caron, L. et al. The causes of sea-level rise since 1900. *Nature* **584**, 393–397 (2020). <https://doi.org/10.1038/s41586-020-2591-3>)
- Llovel, W., Purkey, S., Meyssignac, B. et al. Global ocean freshening, ocean mass increase and global mean sea level rise over 2005–2015. *Sci Rep* **9**, 17717 (2019). <https://doi.org/10.1038/s41598-019-54239-2>

Referee #2 (Remarks to the Author):

Review of Horhold et al: Exceptional temperatures in central-north Greenland ice core

Horhold et al. update a north-central Greenland temperature history derived from a composite of ice-core water isotope records. The decade 2001-2011 is found to be exceptionally warm compared to the previous 1000 years. The north-central Greenland ice core composite (NGT-2012) is compared with the Arctic 2k temperature reconstruction, which has a similar pattern of cooling until 1800 and then warming since, with the warmest period being the 2001-2011 decade. The fundamental conclusion is that the arctic-wide warming also affects the highest elevations of the Greenland ice sheet. The authors suggest that the warm temperature are accompanied by enhanced Greenland meltwater runoff.

The Horhold et al. ice core composite is a welcome update and confirms recent warming is affecting all of the ice sheet. By extending NGT from 1995 to 2010, the recent warming is discernible above the variability.

We thank the reviewer for the constructive assessment of our study and suggestions that followed from it. In the revised version among smaller changes, we substantially improved the analysis of the link between NGT-2012 temperature and Greenland melt, analyzed the relation of our reconstruction to the Greenland Blocking Index and deepened the discussion with respect to Arctic 2k. We also added the accumulation reconstruction to the Extended Figures. In particular, we appreciate the opportunity to present in more detail the statistical relationship between the high-elevation temperatures recorded by NGT-2012 and the meltwater runoff in the ablation zone. With this analysis we support the feasibility of NGT-2012 as a proxy for melt production.

The revision strongly improved the manuscript and show the unique information contained in the presented NGT-2012 data set. For the additional analysis undertaken and modifications made to the manuscript, please see the detailed points below. Answers and changes are given in

colored letters. Note, we have numbered the comments in order to refer to respective answers, when useful. References within our answers are listed at the end. References within the revised text (superscript) refer to references of the manuscript and are not listed here additionally.

Comment 1: The broader significance of the NGT-2012, however, is not clear. North-central Greenland is a small geographic area which in itself does not play an outsized role in global climate.

Answer Comment 1:

- 1) As we demonstrate, NGT is representative of Greenland temperatures, (Figure 2, Extended Data Figure 2, 5) and can be used as a proxy for Greenland meltwater runoff that allows us to provide a long-term reference for this climatically important variable. As detailed below, we further strengthened this argumentation in the revised version.
- 2) Importantly, NGT-2012 is one of the only ‘single proxy’ based millennial temperature records that covers the last millennial including the recent years without switching from proxy data to instrumental data and which is based on a calibration, independent from the instrumental time-series. This is very important, as it circumvents the potential weaknesses of large-scale temperature reconstructions such as Arctic2K, originating from the screening and instrumental calibration step that have been used by the climate skeptics community to discredit the credibility of statements regarding the unprecedentedness of the recent warming (‘Hockey Stick discussion’). While the methodologies and confidence in large-scale reconstructions have strongly improved in the last decades, parts of the methodologies used are still prone to underestimate of past climate variability outside of the calibration period (e.g. Christiansen, B., & Ljungqvist, F. C., 2019; and Extended Data Fig. 7 in Neukom et al., 2019).

Comment 2: The freshwater contribution of Greenland is of global importance, but the potential contribution from this region is not discussed and is likely to be a small part of the total.

Answer Comment 2:

We agree that physically, the contribution of the study region is only a small part to the total Greenland freshwater contribution. However, as we demonstrate now in more detail, the NGT-2012 stack is a good (and to our knowledge, currently the only) proxy for the past total Greenland freshwater contribution. The NGT-2012 stack is representative for the overall Greenland temperatures (new Figure 2) and is highly correlated to overall Greenland meltwater runoff (new in Figure 4, Methods).

We thus use it now to derive estimates of past meltwater runoff (new Figure 4b). Such reconstruction will enable model approaches to estimate past and future Greenland meltwater runoff (i.e. see: Lenaerts et al., 2015, Fettweis et al., 2013, Nicholas et al., 2012) and provides an additional constraint on the 1,000 years of surface mass balance relevant for ice-dynamic modeling of the ice sheet. These estimates will also allow studying the effect of freshwater flux into the ocean from the Greenland ice sheet, which might have impacts on the ocean circulation and climate variability (Bamber et al., 2012, Bakker et al., 2016).

Changes in the manuscript:

- We included a subfigure to **Figure 1** showing the common time period 1871–2011 CE of MAR3.5.2 and NGT-2012 to compare the Greenland meltwater run-off and NGT-2012 temperature
- We re-structured **Figure 4**, showing the meltwater runoff versus NGT-temperature anomaly for both: the common time period 1871–2011 CE, and based on the linear coefficient from the regression, the reconstructed meltwater runoff from the NGT-2012 temperature anomalies
- Main, paragraph **Implications for Greenland’s future meltwater runoff** we added: *“For the period 1871 - 2011 CE we find a strong connection ($R = 0.63$, $p < 0.01$; Methods) between the high-elevation NGT-2012 temperature anomaly and meltwater runoff of the ice sheet.”*

- Main, paragraph **Implications for Greenland’s future meltwater runoff** we added: *“The strong statistical and physically meaningful relationship between the NGT-2012 record and the meltwater runoff allows us to generate the first reconstruction of the meltwater runoff anomalies for Greenland over the last millennium and thus to put the recent runoff anomalies in the long-term context (Figure 4b, Methods). This shows that the meltwater runoff anomalies of the 2001 - 2011 decade are outside the reconstructed distribution of pre-industrial (1000 - 1800 CE) values taking into account the reconstruction uncertainties in our linear model. Therefore, even though with less certainty than for the temperature, our analysis suggests that current decadal meltwater runoff anomalies are unprecedented over the last millennium.”*
- Removed: We compare the NGT-2012 temperature anomalies with Greenland meltwater runoff anomalies from MARv3.12 (Figure 4). We find that the NGT-2012 pre-industrial (1000–1800 CE) temperature anomalies are largely congruent with MARv3.12 for the 1961–1990 CE reference period, commonly associated with a near-balance state (close to zero net mass loss) of the ice sheet⁴⁴. For the most recent decade sampled by NGT-2012 (2001–2011 CE), the temperature anomaly falls within those of the years associated with substantial mass loss in Greenland, including the record meltwater runoff in 2012. While none of the 30 years in the reference period show a meltwater runoff anomaly below - 100 Gt yr⁻¹, this threshold is exceeded for 14 out of 21 years after 2000 CE, or eight out of twelve years in the 2000–2011 CE time period of NGT-2012.
- **Methods:** We rephrase the previous section on MAR to: “MAR3.5.2 surface mass balance (SMB): MAR3.5.2 surface mass balance (SMB) and temperature estimates: *“Melt water runoff is obtained as component of the surface-mass balance (SMB) output of the regional climate model MAR3.5.2 (Modèle Atmosphérique Régional; version 3.5.2²¹). Meltwater runoff refers to meltwater production minus meltwater refreezing, deposition and retention. The MAR3.5.2 simulation used here is forced in six-hourly intervals at its lateral boundaries with Twentieth Century Reanalysis version 2 (20CRv2⁶⁸) for the period 1871–2012 CE, and provides 20 km horizontal resolution. This model output is part of a larger number of 20th century reconstructions of the Greenland Ice Sheet SMB with MAR3.5.2,*

forced by a variety of different atmospheric reanalysis data sets²¹. The 20CRv2 forcing is the ensemble mean of a 56 member experimental reanalysis with spatial resolution of 2.0°, assimilating only surface pressure, monthly sea surface temperature and sea ice cover²¹.

For the period 1980–2010 CE, MAR3.5.2 forced by 20CRv2 has been shown to exhibit a warm temperature bias ($\sim 1^\circ\text{C}$) compared to simulations driven by ECMWF Interim Re-Analysis⁷⁰. However, for the annual melt runoff anomalies with respect to 1961–1990 CE considered in this study, we find that MAR3.5.2 / 20CRv2 is in good agreement with the latest version MAR3.12 forced by the latest reanalysis (e.g., ERA5⁷¹, Extended Data Figure 7), within the common period 1950–2012 CE. Even though it is not possible to measure mass changes due to meltwater runoff directly with satellites, we estimate the meltwater runoff anomaly by subtracting net accumulation (snowfall minus sublimation and evaporation) obtained from MAR3.12 / ERA5 and ice dynamic discharge obtained from InSAR^{41,72} from the GRACE/GRACE-FO annual mass balance with breakpoint January of each year. The results show that the annual variation of the mass budget based on MAR3.12 / ERA-5 is consistent with GRACE/GRACE-FO, as is the budget when replacing the melt runoff from MAR3.12 / ERA5 with the MAR3.5.2 / 20CRv2 estimates (Extended Data Figure 7).

For our study, we base the Greenland meltwater runoff anomalies and 2m surface air temperature data on monthly estimates from MAR3.5.2. The monthly temperature data are sampled at the grid cells closest to the NGT-2012 ice-core locations, averaged across these cells and then averaged to annual mean values; the meltwater runoff data are integrated over the contiguous ice sheet and then cumulated to annual values. Anomalies are calculated with respect to the reference period 1961–1990 CE which is firstly the commonly used reference period in mass balance studies of the Greenland Ice Sheet⁷³ and secondly synchronous to the one used for the NGT-2012 and Arctic 2k time series. Finally, the same 11-yr running mean filter is applied to the annual temperature values as to the NGT-2012 isotope record over the common time period 1871–2011 CE of $R = 0.77$ ($p <$

0.01). Likewise the correlation of the filtered MAR3.5.2 meltwater runoff anomaly with NGT-2012 is $R = 0.63$ ($p = 0.005$)."

- **Methods:** we add a paragraph on the temperature-meltwater runoff relationship: Reconstruction of pre-industrial of melt runoff distribution: "We reconstruct the pre-industrial (1000–1800 CE) distribution of melt runoff anomalies from the pre-industrial NGT-2012 temperature values based on the observed linear relationship between the NGT-2012 temperatures (T_{core}) and MAR3.5.2 melt runoff (M_{MAR}) anomalies for the period 1871–2012 CE,

$$M_{MAR}^{1871-2012} = T_{core}^{1871-2012} \times \beta + \epsilon,$$

where β is the linear regression coefficient and ϵ represents the residual variability not explained by the linear relationship. We estimate $\hat{\beta}$ and its variance $var(\hat{\beta})$ using least-squares adjustment, with the assumption of uniform uncertainties in $M_{MAR}^{1871-2012}$. The reconstructed melt runoff \hat{M} based on pre-industrial T_{core} is then obtained as

$$\hat{M} = T_{core} \times \hat{\beta}.$$

To allow for uncertainties in the relation, represented by both uncertainties in the parameter estimate as well as in the post-fit residual, we calculate the variance of the melt runoff reconstruction

$$var(\hat{M}) = var(M_{MAR}^{1871-2012} - T_{core}^{1871-2012} \times \hat{\beta}) + var(\hat{\beta}) \times T_{core},$$

using a Monte Carlo approach involving 10,000 random samples. To derive the two-dimensional distribution of pre-industrial melt runoff versus temperature data, we create a 2D grid with 50 bins in each direction spanning the range $[T_1, T_2]$ and $\hat{\beta} \times [T_1, T_2]$, where $T_1 = -4$ °C and $T_2 = 4$ °C, and count the number of realizations that fall into each of the bins. The finding of the 2001-2011 decade being outside pre-industrial distribution may seem trivial given the linear dependence on the NGT-2012 data, where the 2001-2011 decade is exceptional. However, as we do consider the uncertainties of the reconstruction, an exceptional NGT-2012 data does not directly imply a statistically significant exceptional runoff. While at a single site, the relation of local temperature and local melt and subsequent runoff is clearly nonlinear, Greenland wide, the warming conditions change

the area affected by melt. Thus, it supports our approximation of the linear runoff response across the range of temperatures experienced in the last millennium.”

Comment 3: It is already widely accepted that the higher elevations in Greenland have experienced recent warming – e.g. satellite observations of melt area (e.g. Colosio et al., 2021), melt layers (e.g. MacFerrin et al., 2019), and regional climate modeling (e.g. Cullather et al., 2016). Thus, the value of the NGT-2012 seems to be less in the decade-old warming signal of 2001-2010 and more in the ability to place that warming signal in longer context.

Answer Comment 3:

We agree that warming has been observed at Greenland coastal areas since the beginning of the 21st century. Alongside with the observed warming, the studies named by the reviewer show trends on surface melt (Colosio et al. 2021, with largest trends in western Greenland, running until 2019), the spatial evolution of refrozen melt layers and their projections (Mac Ferrin et al., 2019) or assess reanalysis and model data on melt over Greenland (Cullather et al., 2016).

We note that none of these studies do address or show if and how central Greenland temperatures are increasing with respect to long-term context which is relevant to better separate natural variability from the human influence. Apart from observations through weather stations, etc., it is most striking that previous firn and ice core studies were not able to detect a signature of warming.

Comment 4: Unfortunately, the authors do not establish a quantitative relationship between the NGT-2012 mean annual temperature (at decadal timescales) and either Greenland mass balance or atmospheric circulation patterns.

Answer Comment 4:

As detailed in the answer to Comment 2, we now establish a quantitative relationship between the NGT-2012 and Greenland meltwater runoff that forms an important part of the mass balance. We further added the analysis on the relationship between NGT-2012 mean annual temperature

and the atmospheric circulation pattern, represented by the Greenland-Blocking-Index (GBI). Our new analysis demonstrates that the annual GBI is significantly correlated to the NGT-temperatures and the summer GBI is significantly correlated to the meltwater-runoff (Delhasse et al., 2021). The analysis suggests that atmospheric blocking contributes to the meltwater runoff (in addition to other parameters such as cloudiness, temperature etc.).

Changes in the manuscript:

- Main, paragraph (now renamed to) **Greenland and Arctic wide temperatures**, we added/rephrased: *“As greater geopotential heights are thermodynamically linked to higher temperatures²⁹, prolonged (...)”*
We added/rephrased: *“(…), and accordingly to increased temperatures on the ice sheet^{28,35,36}. In support of this, we find a significant correlation ($R = 0.63$, $p < 0.005$) between the NGT-2012 temperature record and the Greenland Blocking Index (GBI²⁸; Extended Data Figure 2d, e), supporting Greenland blocking as one reason for the larger variability at decadal time scales of the NGT-2012 record compared to Arctic 2k. Greenland blocking was suggested to influence surface melt by influencing the advection of warm air masses^{35,36}. Indeed, we find a high correlation between GBI and the Greenland meltwater runoff, derived from the regional climate model MAR3.5.2 ($R = 0.80$, $p < 0.001$, Methods, Extended Data Figure 2f). During the past decades, the GBI increased in frequency, and to some extent, persistence and magnitude, particularly in summer³⁶. This indicates that blocking conditions³³⁻³⁵, superimposed on thermodynamic warming and natural decadal temperature variability, have contributed to the observed records of summer melt in Greenland.”*
- **Methods**, we added a paragraph on GBI correlation analysis: “Comparison with Greenland Blocking Index: *“We compare the Greenland Blocking Index GBI²⁸ time series to the NGT-2012 temperature and MAR3.5.2 meltwater runoff data over their common time periods. Using 11-year running mean filtered data, the correlation between NGT-2012 and annual GBI is $R = 0.63$ ($p = 0.002$) and between meltwater runoff and annual GBI it is $R = 0.80$ ($p \ll 0.01$). Replacing the annual GBI data with the average GBI for summer*

(months JJA), the correlation with meltwater runoff is $R = 0.91$ ($p < 0.01$). The correlations are robust also for the unfiltered annual mean values, with correlations of $R = 0.38$ ($p < 0.01$), $R = 0.56$ ($p < 0.01$), and $R = 0.67$ ($p < 0.01$), respectively.”

- **Extended Data**, we restructured **Extended Data Figure 2**, showing now the comparison of NGT-2012 with instrumental and model data: we added as subfigure (2d): NGT-2012 anomaly and GBI over the 1871-20211 CE period, and (2e) scatter plot of NGT-2012 and GBI as well as (2f) meltwater runoff and GBI.

The manuscript has at least three notable gaps.

Comment 5: First, only the annual water isotope records are used. The cores are annually dated and have density profiles, so the annual accumulation rate is also known. Accumulation rates – effectively precipitation in this region of the ice sheet – are a fundamental component of the ice sheet’s contribution of sea level.

Answer Comment 5:

We agree that accumulation rate reconstructions provide valuable information on the surface mass balance of the ice sheet and thus to the overall ice sheet's mass balance.

Unfortunately, the accumulation rate records from the NGT cores are characterized by a low signal to noise ratio. This is consistent to earlier studies describing the high temporal and spatial variability of the accumulation rates in this region. As the high noise level does not allow robust inferences on the temperature to accumulation relationship or trends in accumulation, we added the results on the accumulation rate reconstruction in the Extended Data section as additional information.

In detail:

Analyzing the coherency of the accumulation records, we find a signal to ratio (SNR) of the mean NGT accumulation time-series that is much lower than the SNR of the isotope record. For time-scales less than centuries, the estimated SNR is ≤ 1 (Figure R1a). One reason is the strong spatial

variability in local accumulation rates (e.g. Vallelonga et al., 2014 for the NEGIS ice core site, Karlsson et al., 2016 for the NEEM-NGRIP line). This affects the accumulation rate reconstructions as local noise (as one core is not representative of a region, Mosley-Thompson et al., 2001, Zühr et al., 2021) and can also create long-term artifacts if the spatial variability upstream the core-site affects the down-core record by ice-flow. (Hawley et al., 2014, Vallelonga et al., 2014).

As expected from the low SNR, the correlation between the NGT-2012 mean accumulation rate and d18O is weak ($R = 0.23$, $p = 0.05$) (Methods, Extended Data Figure 6). This weak correlation between isotopes and accumulation rate in North-East Greenland and/or high elevation sites is consistent to earlier findings; Berkelhammer et al., (2016), found an increase (decrease) in summer (winter) in accumulation but argued that the response of accumulation rate to temperature is highly variable due to decoupling of surface and atmosphere (Furukawa et al., 2017). Buchardt et al., (2012), compiled approx. 50 ice core records to study the link between accumulation rate and stable water isotopes record. They found an overall weak correlation and a region-specific temperature to isotope dependency, which is consistent to our findings (Figure R1b).

The reconstructed accumulation rate shows an increase in parallel with the d18O signal since the 2000 AD (as observed also in e.g. Haris and Simons 2012), but the shortness of the trend and the missing relationship on the earlier time-period does not allow to make robust inferences on the temperature to accumulation rate relationship. This is consistent with earlier studies; e.g. Buchardt et al., 2012 (from firn cores) and Kjaer et al., preprint (from radar) looking at the ice divide and Central to North-East Greenland did not report a statistically significant trend in recent accumulation rates.

As the accumulation rate reconstructions are only weakly related to our main study results and do not contribute a major new finding relative to the existing literature, we describe these results in the Extended Data section and will provide our accumulation reconstruction in PANGAEA for the use in studies focused on accumulation variability.

Figure R1: The NGT-2012 accumulation rate. **a)** Signal-to-Noise ratio of the NGT-2012 accumulation rate and **b)** NGT-2012 accumulation rate anomaly vs NGT-2012 isotope anomaly together with the regressions for North-East (NE) and North-West (NW) Greenland derived by Buchardt et al., (2012).

Changes in the manuscript:

- We added accumulation rate to the data sets of B18-2012, B21-2012, B23-2012, B26-2012 and NGRIP-2012 (**Pangaea**)
- Main, paragraph **Implications for Greenland’s future meltwater runoff**, we added: *“In principle, the higher meltwater runoff could be partly compensated by an increase of accumulation accompanying the warmer temperatures. While accumulation reconstructions from the NGT-2012 stack are much more uncertain than the NGT-2012 based temperature reconstruction (Methods), they do not provide evidence for a strong link of temperature and accumulation or unprecedented accumulation in the last decade (Extended Data Figure 6).”*
- **Methods**, we added one sentence on accumulation rate in the data set description: *“Accumulation rates for the extension cores were derived from the density measurements and the depth-age profiles obtained from the annual layer counting.”*

- **Methods**, we added a paragraph on the accumulation rate stack: “The NGT-2012 accumulation rate stack:” *We compile a simple stack from the single accumulation rate records (Extended Data Figure 6a), for which we use the data from the extension cores as well as from B16, B18, B21, B26, B29 and NEEM; the data of the remaining cores were not used due to insufficient quality. Analyzing the coherency of the accumulation records, we find a signal to ratio (SNR) of the mean NGT accumulation time-series that is much lower than the SNR of the isotope record. For time scales less than centuries, the estimated SNR is ≤ 1 record (Extended Data Figure 6b). For the stacked record, we here use a simple stack and do not merge first the available pairs of old and extension records, as it is done with the isotope data, as such an extension of the single was not applicable due to the much higher noise level of the accumulation records. One reason is the strong spatial variability in local accumulation rates. This affects the accumulation rate reconstructions as local noise (as one core is not representative of a region) and can also create long-term artifacts if the spatial variability upstream the core-site affects the down-core record by ice-flow. As expected from the low SNR, the NGT-2012 isotope and accumulation rate data exhibit a low correlation of $R = 0.23$ ($p = 0.05$) over 1500-2011 CE, without any statistically significant linear relationship. Even though the accumulation rate has been increasing since 2000 CE, similar to isotopic data, this time interval is too short to derive any general relationship. In addition, the 2001 - 2011 CE block averaged accumulation rate is not exceptional in the context of the pre-industrial values (Extended Data Figure 6c) which could be due to noise in the reconstruction or a low sensitivity of the accumulation on the recent climate change.”*
- We added **Extended Data Figure 6**. NGT-2012 accumulation rate data, showing the time 1500-2011 CE time series of both stacks (isotopes and accumulation rate), the signal-to-noise ratio over time-periods and the probability density distribution

Comment 6: Melt layers have also been used extensively as a relative proxy of summer melt. While I’m not familiar with many of these core sites, NEEM has well documented ice layers and thus many of the cores likely have sufficient melt to have preserve melt layers.

Answer Comment 6:

We agree that studying the occurrence, frequency and spatial extent of melt from our ice cores is a highly interesting approach. The first melt record for North-East Greenland was established from the Recap ice core retrieved from the near-coastal ice cap Renland (Taranczewski et al., 2015), showing the history of frequency and intensity of melt throughout the past 100 to 10.000 years. The NEEM melt record (Orsi et al., 2015) served as a basis to study the effect of bubble free (i.e. refrozen melt) layers on the air content in the ice (and thus on the concentration of gases), looking at the previous warm period/interglacial, the Eemian. Similar studies from Greenland deep ice cores look at melt events in the early Holocene (Alley et al., 1995, Westhoff et al., 2022).

Nevertheless, in Central-North Greenland surface melt is a very rare feature. This is one reason why the 2012 melt event (Nghiem et al, 2012) has caused so much attention, because it was the first observed Greenland-wide (i.e. also at the high elevation, central-north Greenland) event of this kind. The only other melt layer seen in many ice and firn cores from Central-North Greenland, i.e. being of a similar extreme nature, dates back to 1889 (Keegan et al., 2014, Orsi et al., 2015). Thus, while we do find few scattered melt layers in some of the cores (depending on the specific position) the presented NGT-2012 cores were still too cold and thus do not have sufficient melt to have preserved melt layers for an in-depth study. Our firn cores were drilled in spring 2012 (2011), just before the 2012 event happened (followed by more in the years after). So, while we agree on this suggested approach, we have to postpone it to the next generation of NGT-extensions, when the most recent decade (after 2012) is covered and potentially many of the NGT-sites do see an increasing number of melt events that could be placed into the long-term context and compared to the expectations from the isotope record.

Comment 7: Both accumulation rates and melt layers are also useful for interpreting atmospheric circulation patterns; annual accumulation rates because they have shorter correlation length scales than temperature records and melt layers because they are primarily influenced by summer temperature.

Answer Comment 7:

As detailed in Comment 4, we now analyze the relationship of NGT-2012 to atmospheric circulation patterns, by making use of the Greenland Blocking index (GBI). We find a high significant correlation between GBI and our annual mean temperature as well as with the meltwater runoff.

With respect to accumulation rates and melt layers: As we detailed above, NGT-2012 firn cores unfortunately do not provide melt layer records as until 2012 widespread melt events affecting most of our firn core locations did not take place (see answers Comment 6). For our accumulation rate records we show that the Signal-To-Noise ratio is much lower than for the isotope record (please see Comment 5), we therefore do not elaborate on potential correlation analysis of accumulation rate and e.g. GBI.

Changes in the Manuscript:

Please refer to changes as in answer to comment 4

Comment 8: There is a suggestion that the warm air temperatures are due to atmospheric blocking but the authors provide no new analysis and insight on this potential mechanism for enhanced melt.

Answer Comment 8:

We added a new analysis studying a) the relationship of NGT-2012 temperature and Greenland meltwater runoff, b) NGT-2012 temperature and Greenland blocking index GBI and c) we show the relationship between Greenland meltwater runoff with the blocking index GBI and now include a discussion on potential mechanisms.

Changes in the Manuscript

Please refer to changes as in answer to comment 4

Comment 9: There is also mention that arctic warming is greatest in the winter and it seems like many of the records in the NGT-2012 stack should have seasonal resolution to address this interesting question.

Answer Comment 9:

While the sampling resolution of our cores is in principle high enough to resolve the seasonal cycle, the diffusion together with the low accumulation rate strongly limits the ability to robustly separate seasonal isotopic variations at our sites. In contrast to the pioneering study of Vinther et al., (2010) of South-West Greenland ice-cores with mid to high accumulation rates (0.22-0.57 m ice/yr) our cores from Northern Greenland are characterized by substantially lower accumulation rates (0.09-0.17 m ice/yr).

This lower accumulation rate results in a substantial difference on the preservation of the seasonal signal after diffusion. While at an accumulation rate of 0.3m ice/yr, the amplitude of the annual cycle at the firn-ice boundary is reduced to 25% and thus can be recovered using deconvolution ("back-diffusion), at an accumulation rate of 0.15m ice/yr the amplitude of the annual cycle is reduced to 0.3% (assuming 8cm firn diffusion length) and can't be recovered. The deconvolution would also amplify the measurement noise by the same amount (a factor of 300).

In addition to this technical limitation for most of our sites; we would not expect a major gain from a seasonal separation as 1.) Annual mean isotope values have a lower SNR than winter-only isotope values (Vinther et al., 2010) and 2.) Winter and annual temperatures are highly related ($R^2 > 0.9$ for SW Greenland, Vinther et al., 2010).

Comment 10: Second, the relationship between NGT-2012 and melt-water runoff is confusing and not developed statistically. One primary limitation is that the NGT-2012 only appears representative on decadal timescales, preventing annual comparisons. But even with the most recent decade (2001-2010), it is not clear that the MARv3.12 2m air temperature anomalies agree very well with NGT-2012 ones. In Figure 4, the only years I see to the right on the 2001-2010 NGT-

2012 average are 2010 and 2003 (just barely). Which means the MARv3.12 decadal average must be considerably cooler.

Answer Comment 10:

We have substantially improved the analysis of the relationship between NGT-2012 temperature anomaly and meltwater runoff by extending the comparison to a longer time-period and show that the results are also robust on annual time-scales. For this we now use MAR3.5.2 20th century reconstructions of the Greenland SMB (1871-2011) forced by climate reanalysis. Comparison with MAR3.12 forced by ERA5 and comparison with GRACE/GRACE-FO indicates that MAR3.5.2 provides a good reconstruction of SMB (Extended Data Figure 7).

The correlation is also statistically highly significant on annual data, showing that the NGT-2012 record is also representative on faster than decadal time scales (e.g. Extended Data Figure 5, Methods) and that our choice of focusing on decadal mean data (based on the SNR estimates) is conservative.

Further, we do find a good correlation between MAR3.5.2 temperature and NGT-2012 temperature of $R = 0.77$ ($p < 0.001$) (Extended Data Figure 2b).

Changes in the manuscript:

Please refer to changes in answer to comment 2

Comment 11: But maybe the larger question is whether annual temperature is even a particularly useful metric for runoff which is largely controlled by summer (melt season) temperature. Thus, it is not clear that the NGT-2012 record is well suited for assessing Greenland melt water runoff.

Answer Comment 11:

We agree that the underlying physical relationship is between melt-season temperature and runoff. However, our stacked annual mean temperature reconstruction is a good proxy for the runoff as the interannual and longer temperature changes are highly correlated between the different seasons. The (annual) NGT-temperature record is strongly related ($R = 0.63$, $p = 0.005$)

to the meltwater runoff from the regional climate model MAR (Figure 1, Methods) supported by satellite observations (Extended Data Figure 7). We note that even if the ice-core data would technically allow a seasonal separation of water isotopes (see our Answer to Comment 9) it is unclear if summer isotope values would be a better proxy for runoff. Vinther et al. (2010) found that the annual mean water isotope signal is an equally good or better proxy for summer temperatures than the summer isotope signal (Fig. 3a, b in Vinther et al., 2010) and also showed that the annual mean water isotope signal has the highest signal to noise ratio.

Comment 12: Third, the NGT-2012 is only compared to Arctic2k, but not any other nearby climate histories. Arctic2k is composed of many records that could be grouped regionally. In particular, there are surrounding ice core records from other locations in Greenland and the Canadian and European Arctic.

Answer Comment 12: Please refer to answer to comments 13 and 21

Comment 13: The nearly opposite spatial correlation maps between NGT-2012 and Arctic2k with 20CR deserves considerably more attention. The greatest concentration of records in Arctic 2k is actually from the same region as NGT-2012. The authors note that these records must not be contributing significantly to Arctic 2k because of the low correlations, but the cause of the lack of influence is not explained. The NGT-2012 and Arctic 2k show the same overall pattern for the last millennium (cooling until ~1800 and then warming), **so the question of what timescales the two are related on should be explored more.**

Answer Comment 13:

We deepened the analysis of the complementary spatial correlation maps of NGT-2012 vs. Arctic 2K and the relation of the two reconstructions as a function of timescale.

To assist the interpretation of the spatial correlation maps, we reproduced the correlation maps after replacing the imperfect reconstructions of NGT and Arctic 2K by the temperature time-

series of the region extracted from the reanalysis temperature data (Extended Figure 5). This results in virtually the same pattern demonstrating that this is not an artefact of the reconstructions

A coherency analysis of the two time-series (now added to Figure 3) reveals that they are only related on multi-decadal and longer time-scales. To exclude, that this is just a result of the imperfect NGT reconstruction, we also estimated the coherency between the NGT stack and the temperatures averaged in the NGT region derived from the 20CR reanalysis data. This confirms that the relation of NGT and local temperatures is already high and significant at longer than decadal time-scales; consistent with our choice of 11 year means as robust temperature signal.

Therefore, the new coherency analysis, together with the difference in the power spectral densities (also Figure 3) suggests that the individual variability of Greenland and Arctic2K in the decadal-to multi-decadal time periods is a climatic feature. This is also consistent to earlier findings (Arctic 2K, Figure S2, Supplementary information) of decadal scale anti-phasing between Greenland and Fennoscandia.

Finally comparing the NGT-2012 stack with all individual annually resolving records from Arctic 2k and comparing the Arctic 2k temperature reconstruction with the individual proxy records also confirms the separation of the Greenlandic variability for the last centuries beyond the instrumental period (Figure R3). This separation is weaker as the long time series also includes contributions from the longer time-scales (multi-decadal and more) in which the NGT-2012 and Arctic 2k are coherent but the general pattern is also supported on these long proxy time-series. As this analysis is only indicative as the different proxy types are subject to different noise levels and time-scale dependent biases we didn't include it in the revised manuscript.

Changes in the Manuscript:

- **Figure 3:** we added the coherence of NGT-2012 and Arctic 2k as well as NGT-2012 and 20CRv3

- **Main**, paragraph **NGT-2012** we added: *“Throughout the past millennium, our ice-core based Greenland temperature reconstruction and the Arctic wide temperature reconstruction are correlated ($R = 0.66$, $p < 0.01$, 1000–2011 CE), but this correlation does not persist when limiting the comparison to the 20th century ($R = 0.29$, $p = 0.16$; Methods; Extended Data Figure 4) the time-period that arguably has the best reconstruction quality.”*
- **Main**, paragraph (now renamed to) **Greenland and Arctic wide temperatures** we added: *“In a first look, this is surprising as a number of Greenland ice-core records are included in the reconstruction.”*
- **Main**, paragraph (now renamed to) **Greenland and Arctic wide temperatures** we added: *“(…) - a result which is also robust for annual mean values that are subject to more reconstruction uncertainty (Extended Data Figure 5). The distinct spatial correlation structure is not an artefact of the reconstructions. Replacing the NGT-2012 reconstruction by the temperature extracted from the 20CR reanalysis for the region represented by the NGT-2012 and Arctic 2K results in virtually the same complementary patterns (Extended Figure 5).”*
- **Main**, paragraph (now renamed to) **Greenland and Arctic wide temperatures** we added: *“This implies that one single time-series alone does not provide a good representation of the Arctic temperature evolution. Here, our Greenland reconstruction and Arctic 2k together provide a more complete picture in the assessment of past and recent temperature changes in the circum-Arctic region and are an important step towards spatio-temporal reconstructions of the Arctic temperature evolution.” “The decoupling is also visible in the distinct spectrum of temperature variability. (…)”*
- **Main**, paragraph (now renamed to) **Greenland and Arctic wide temperatures** we added: *“At the same time, both temperature reconstructions exhibit similar power spectral densities for time periods above 50 years and below 8 years. This indicates that the different spatial coverage of the reconstructions is not the primary reason for the variability difference for decadal to multi-decadal time periods as a different spatial coverage is expected to mainly affect the short time-scales. Analyzing the relation of the*

Greenland and Arctic reconstructions shows a high coherence at time periods longer than 50 years, which, however, drops towards shorter time periods (Figure 3 lower panel). In comparison, the coherence between NGT-2012 and the local temperature (NGT@20CRv3) remains high down to time periods below 20 years. This demonstrates that the decoupling between NGT-2012 and Arctic 2k on the decadal to multi-decadal time-scales is not an artefact of the NGT reconstruction quality.”

- **Methods**, we added in paragraph **Spectral Analysis**: “We estimate the squared coherency between time series to assess their linear relationship as a function of timescale using the smoothed periodogram. Confidence levels are obtained by replacing the original time series with AR1 red-noise surrogate time series with the same autocorrelation and using the frequency averaged $p = 0.95$ sample quantile of $n = 1,000$ realizations.”

Figure R3: Correlation of temperature reconstructions with the individual proxy records used in Arctic 2k. a) NGT-2012 vs. the Arctic 2k proxy records; b) Arctic 2k reconstruction vs. Arctic 2k proxy records. Correlations of 11yr running mean time-series are shown. Crosses mark statistically non-significant correlations ($p > 0.05$). The NGT-2012 stack is mainly correlated to sites from Greenland, whereas the Arctic 2k reconstruction shows stronger correlations outside of Greenland.

Comment 14: The statistics of the NGT-2012 are, overall, well presented. There are some minor issues that could be addressed, like including a subfigure of the number of records at any given time. This information is in the Extended Data Table 1, but is not easy to discern.

Answer Comment 14:

The number of records included in the NGT-2012 stack was shown in the previous Extended Data Figure 5b. We have now moved it to Figure 1

Changes in the manuscript:

- Previous Extended Data Figure 5: remove number of records
- Figure 1: Include number of records

Comment 15: Where the statistical analysis seems to fall apart is with the relationship to Greenland surface melt – there does not seem to be any statistical analysis of the relationship between NGT-2012 and MARv3.12 runoff.

Answer Comment 15:

We agree that the analysis on the link between meltwater run-off and temperature was not presented in-depth in the manuscript. We have refined our analysis: We now provide a statistical analysis of the relationship of meltwater runoff of the entire ice sheet and NGT-derived temperatures and are able to show that for the 1871 to 2011 period the NGT-temperature record is significantly highly correlated to the meltwater runoff (Figure 1, Methods).

Changes in the manuscript:

Please see changes as in answers to Comment 2 and the respective changes in the manuscript.

Comment 16: I expected a quantitative relationship to be established between the two such that a record of melt could be extended for the past millennium with NGT-2012. I imagine this was

not possible given the lack of annual reliability in the NGT-2012, but without a quantitative relationship, the importance of NGT-2012 on Greenland melt remains speculative.

Answer Comment 16:

We now established the quantitative relationship that also allows us to extend the meltwater runoff record back in time. We added the reconstructed meltwater runoff anomalies from the NGT-temperature reconstruction of the pre-industrial period (new in Figure 4). Such reconstruction will enable model approaches to estimate past and future Greenland meltwater runoff (i.e. see: Lenaerts et al., 2015, Fettweis et al., 2013, Nicholas et al., 2012) and provides an additional constraint on the 1,000 years of surface mass balance relevant for ice-dynamic modeling of the ice sheet. These estimates will also allow study the effect of the freshwater flux into the ocean from the Greenland ice sheet, on the ocean circulation and climate variability; (Bamber et al., 2012, Bakker et. al., 2016). We also a find a statistically significant relationship on the annual basis (NGT-2012 to meltwater runoff, $R = 0.41$, $p < 0.01$. NGT-2012 and GBI: $R = 0.38$, $p < 0.01$) and thus our choice of a decadal scale analysis (based on the SNR estimate) is conservative.

Changes in the Manuscript:

Please refer to Comment 2 and the respective changes in the manuscript.

Comment 17: Another broad comment is that the figures are not effective. Figure 3 could easily be incorporated into Figure 1, and Figure 2 seems more appropriate for the supplement (or Figure 1 as well). This leaves two full figures that could be dedicated to more detailed analysis.

Answer Comment 17:

We improved the composition of the figures to include more detailed analyses in the main manuscript

Changes in the manuscript

- **Figure 1** includes the two time series Arctic 2k and NGT-2012 for their common period, the number of cores through time from the NGT-2012 record, the map of NGT-2012 sites and the 1871-2011 time series of meltwater runoff from MAR together with NGT-2012
- **Figure 2** shows the spatial correlation of NGT-2012 and Arctic 2k with 20C reanalysis data (from previous Extended Data Figure 3a and b)
- **Figure 3** shows the NGT-Artic2k relationship with the a) the variability in the frequency domain (previous Figure 2), and b) the coherence analysis of the NGT-2012 and Arctic 2k and NGT-2012 and 20CRv3 time series
- **Figure 4a** shows the Probability density (previous Figure 3) of NGT-2012 pre-industrial distribution together with the 2001-2011 block average and **Figure 4b** shows the MAR meltwater runoff anomaly versus NGT-2012 temperature anomaly for 1) the 1871-2011 common time period and 2) the reconstructed meltwater runoff for the pre-industrial period. The 2001-2011 block average is shown as well.

Specific comments:

Comment 18: L79 – Why are cores from other areas not included if the goal is to “reconstruct and analyze the Greenland temperature evolution over the last decades with respect to natural variability and global warming”. Many other ice core records exist, some of which have been updated like Renland ice cap.

Answer Comment 18:

The comment raises a good and challenging point, i.e. which ice cores to include in order to reconstruct a regional climate signal.

Such a choice is a trade-off between limiting the analysis to cores from one region in order to maximize a coherent climate signal and to including cores from other parts of the ice-sheet in order to maximize the spatial coverage (see also answer to comment 21).

The aim of the NGT-2012, was to create a temperature reconstruction for central and north Greenland, representing the high-altitude region of the ice-sheet where one would expect the least, or latest changes with respect to global warming. We included a few records from known and established deep ice core sites of this region (e.g. NEEM, GISP2, GRIP, NGRIP and NEGIS).

Potential ice cores that one could consider to add are a) ice cores from south-west Greenland and b) coastal ice core such as from the Renland ice cap. For a) studies looking at shorter time scales report that Northern-Greenland temperature reconstructions do not agree/represent southern Greenland (i.e. Dye 3) climate well (Badgley et al., 2020), even though they do agree on longer (e.g. longer than centennial) time scales (Johnsen et al., 2001, Vinther 2011). For b) we note that coastal ice core sites are prone to varying temperature-d18O relationships, with the d18O variation being sensitive to hydrology, impact of nearby sea-ice conditions, varying moisture sources etc. This is shown for the Renland ice core (Home et al., 2019) and also discussed for Western Greenland in Cluett et al., (2021), but also for other coastal sites e.g. in Antarctica (Breant et al., 2019).

Thus, we decided to keep the composition of the NGT-2012 stack as a record for central-north Greenland (that statistically shows a spatial correlation over the whole Greenland ice sheet with 20 Century Reanalysis Data; Figure 2 and Extended Data Figure 7). but made phrasing in the text more precise with respect to the regional coverage.

Other choices might be possible and given the heterogeneous temperature evolution of the Arctic and Greenland it might be worth to attempt an Arctic wide temperature field reconstruction using Greenland ice-cores beyond the NGT-2012 stack as well as the Arctic 2K records. However, this is beyond this study.

Changes in the manuscript:

- Abstract, Line 20: Replaced: “... also arrived in central Greenland” to “... also arrived in central-north Greenland”
- Main, Line 47: Replaced “central” with “central-north”
- Main, Line 253: Replace “on top of the Greenland ice sheet” with “in central-north Greenland”

Comment 19: L86 – “unprecedented” – wasn’t there an NGT-1995 which covered this time period already? So unprecedented does not seem accurate.

Answer Comment 19:

The (previous) line 86 states that the NGT-2012 record covers more than 1000 years, making it unprecedented in its spatial and temporal coverage of the area. The “NGT-1995” stack (Weißbach et al., 2016) covers the time period until the year 1995. As detailed in the Methods we here extend this NGT-1995 record both in space (i.e. adding NEEM, NEGIS) and time (i.e. by the extension to the year 2011 of the NGT cores B18, 21, 23, B26, NGRIP). Thus, the NGT-2012 covers a larger spatial area and larger temporal range than the NGT-1995 and is thus “unprecedented” in this respect.

Comment 20: Figure 3 – This is a clear figure but it would be more effective in Figure 1.

Answer comment 20:

The previous figure 3 is extended by a coherence plot, to address the relationship between NGT-2012 and Arctic 2k in more detail. Figure 1 is now extended by the number of firn cores and, for the coeval time period, with MAR meltwater runoff and NGT-2012. See also comment 17

Comment 21: L185 – Arctic 2k uses many of the NGT ice core records. In fact, north-central Greenland has among the greatest density of climate records. How is NGT-2012 filling a spatial gap? Does NGT-2012 changes anything from 1000-1995?

Answer Comment 21:

Both reconstructions indeed share some ice-core records (Table R1). However, NGT-2012 includes ten additional cores not used in Arctic 2k. As Arctic 2k aims to reconstruct the area-mean temperature of the full Arctic (60-90N), it results in a reconstruction that is not representative of Greenland (Figure 2, main text). Thus, already from 1000-1995 the information gained from both reconstructions are complementary. Due to the redrilling performed for this study, only NGT-

2012 allows to robustly set the recent decade in Greenland into the long-term context (single-proxy without switching between proxy and instrumental data).

Our results further imply that one single time-series does not provide a good representation of the Arctic temperature evolution. Here, our Greenland reconstruction and Arctic2K together provide a more complete picture in the assessment of past and recent temperature changes in the Arctic region and are an important step towards spatio-temporal reconstructions of the Arctic temperature evolution.

Ice core	In Arctic2k / last year in the record	In NGT-2012 / last year in the record
Crete	X / 1973	
Camp Century	X/ 1967	
Renland	X / 1980	
GRIP	X / 1979	X/1979
Dye3	X / 1979	
GISP2	X / 1987	X/1987
NGRIP	1/ 1995	X/ 2011
B16	X / 1992	X / 1992
B17		X/ 1992
B18	X/ 1992	X/ 2011
B20		X/ 1993
B21	X / 1993	X / 2011
B22		X/ 1993
B23		X/ 2011
B26		X/ 2010
B27/28		X/1994
B29		X/1994
B30		X/1988
NEEM		X/ 2011

NEGIS		X/ 2011
-------	--	---------

Table R1: Greenland ice core records of the Arctic 2k and NGT stacks

Comment 22: L191 – The stronger temperature variability in NGT-2012 seems like an expected consequence of the small spatial region. What is the expectation for how much greater variability a more geographically limited region should have? Comparing the increased variability to a reasonable expectation would be more informative.

Answer Comment 22:

We agree with the reviewer that in general, more geographically limited regions are expected to have more variability. As the spatial covariance of surface temperature tends to increase on longer time-scales (e.g. Kunz and Laepple 2021) we would expect the largest effect of the spatial scale on the fast frequencies with more local averages showing more variability (Figure R4). To demonstrate this we compare the area mean power spectral density (PSD) of local Arctic spectra (calculated on every grid cell, 60-90N, red) with the spectra of the mean Arctic temperature (black).

As expected, we find the largest difference at the fast frequencies (Figure R4). Deriving a more quantitative expectation is challenging as the amplitude of climate variability is also spatially dependent and the effective region of averaging (sampled by the proxy records) is unclear. However, in contrast to what is expected by the spatial coverage, the spectral power density of the two records NGT-2012 and Arctic 2k show a similarity in time periods shorter than 8 and longer than 50 years (Figure 3, main text). Both do show increased power for decadal variability, with NGT-2012 being larger in magnitude. This demonstrates that our results are not a trivial result from a different spatial coverage but imply more decadal-multi-decadal variability in Greenland than in the Arctic. We thus left our interpretation unchanged.

Figure R4: Spatial average and local temperature spectra of the Arctic surface temperature (60-90N) from 20CR reanalysis. The PSD of local temperatures (red) is higher than the spectra of the mean temperature (black) with the largest difference at the fast frequencies. This contrasts with our result from the NGT and Arctic2K spectra that show similar variability at low and high frequencies.

Changes in the manuscript

- Please refer to comment 13 and the respective changes

Comment 23: L192 – What is the new evidence for atmospheric blocking in 2001-2010 compared to previous decades with reliable climate reanalyses? It seems like this has been well established by others (e.g. Graeter et al., 2018)

Answer Comment 23:

We added the analysis to study the relationship between NGT-2012 mean annual temperature and the atmospheric circulation pattern, represented by the Greenland-Blocking-Index (GBI). Our new analysis demonstrates that the annual GBI is significantly correlated to the NGT-temperatures and the summer GBI is significantly correlated to the meltwater-runoff, as has been

shown in previous studies (e.g. Hahn et al., 2018, Delhasse et al., 2021). Our analysis suggests that atmospheric blocking contributes to the meltwater run-off in addition to other parameters such as cloudiness, temperature etc. Based on the above observations of the linkage of GBI and increased temperature (McLeod et al., 201) the Graeter et al., (2018) paper, which looks at firn cores from the percolation zone in South-West Greenland, i.e. an area, where melt happens on an annual basis, compares their derived melt feature percentage with Western Greenland temperature and the 500mb geopotential height and North Atlantic sea surface temperature from MAR and reanalysis model. In comparison, in our study we explore the link with ice cores from areas further north and from higher elevations, extending the link between GBI and temperature reconstructions from ice cores.

Changes in the Manuscript:

Please refer to comment 4 and respective changes

Comment 24: L246 – Can you quantify the impacts on firn densification and meltwater storage? What are the impacts relative to changes in the rest of Greenland?

Answer comment 24:

To disentangle and quantify these effects is beyond the scope of this study and may be addressed in future. We like to keep the potential effect on firn densification & meltwater storage as a general statement.

Changes in the manuscript:

- Main, Line 231ff was modified to: *“This will likely affect the firn densification and the potential for meltwater storage^{18,26,39,45} with further implications for the ice sheet mass balance.”*
- Main, Line 257ff was modified to: *“(…), firn properties such as permeability and melt water retention may change, comparable to firn changes observed in warmer, and lower elevation areas.”*

References by Reviewer 2:

- Colosio, P, M. Tedesco, R. Ranzi, X. Fettweis, 2021. Surface melting over the Greenland ice sheet derived from enhanced resolution passive microwave brightness temperatures (1979-2019). *The Cryosphere*, 15(6), 2623-2646.
- Cullather, R.I., S.M.J. Nowicki, B. Zhao, L.S. Koenig, 2016. A characterization of Greenland Ice Sheet surface melt and runoff in contemporary reanalyses and a regional climate model, *Frontiers in Earth Science*, 10.3389/feart.2016.00010
- Graeter, K. A., Osterberg, E. C., Ferris, D. G., Hawley, R. L., Marshall, H. P., Lewis, G., et al. (2018). Ice core records of West Greenland melt and climate forcing. *Geophysical Research Letters*, 45, 3164–3172. <https://doi.org>
- MacFerrin M., H. Machguth, D. van As, C. Charalampidis, C. M. Stevens, A. Heilig, B. Vandecrux, P. L. Langen, R. Mottram, X. Fettweis, M. R. van den Broeke, W. T. Pfeffer, M. S. Moussavi & W. Abdalati, 2019. Rapid expansion of Greenland’s low-permeability ice slabs, *Nature*, <https://doi.org/10.1038/s41586-019-1550-3>

References (Answer Review 2)

- Alley, R., & Anandakrishnan, S. (1995). Variations in melt-layer frequency in the GISP2 ice core: Implications for Holocene summer temperatures in central Greenland. *Annals of Glaciology*, 21, 64-70. doi:10.3189/S0260305500015615
- Badgeley, J. A., Steig, E. J., Hakim, G. J., and Fudge, T. J.: Greenland temperature and precipitation over the last 20 000 years using data assimilation, *Clim. Past*, 16, 1325–1346, <https://doi.org/10.5194/cp-16-1325-2020>, 2020.
- Bakker, P., et al. (2016), Fate of the Atlantic Meridional Overturning Circulation: Strong decline under continued warming and Greenland melting, *Geophys. Res. Lett.*, 43, 12,252– 12,260, doi:10.1002/2016GL070457.)
- Bamber, J., van den Broeke, M., Ettema, J., Lenaerts, J., and Rignot, E. (2012), Recent large increases in freshwater fluxes from Greenland into the North Atlantic, *Geophys. Res. Lett.*, 39, L19501, doi:10.1029/2012GL052552.

- Berkelhammer, M., Noone, D.C., Steen-Larsen, H.C., Cox, C.J., O'Neill, M.S., Schneider, D., Steffen K., and J. W. C. White; 2016, Surface-atmosphere decoupling limits accumulation at Summit, Greenland, *Science Advances*, Vol. 2, issue 4, <https://www.science.org/doi/10.1126/sciadv.1501704>
- Camille Bréant, Christophe Leroy dos Santos, Cécile Agosta, Mathieu Casado, Elise Fourré, et al..Coastal water vapor isotopic composition driven by katabatic wind variability in summer at Dumont d'Urville, coastal East Antarctica. *Earth and Planetary Science Letters*, Elsevier, 2019, 514, pp.37-47. 10.1016/j.epsl.2019.03.004(Berkelhammer, Noone et al. 2016)
- Buchardt, S. L., Clausen, H. B., Vinther, B. M., and Dahl-Jensen, D.: Investigating the past and recent $\delta^{18}\text{O}$ -accumulation relationship seen in Greenland ice cores, *Clim. Past*, 8, 2053–2059, <https://doi.org/10.5194/cp-8-2053-2012>, 2012.
- Cluett, A. A., Thomas, E. K., Evans, S. M., & Keys, P. W. (2021). Seasonal variations in moisture origin explain spatial contrast in precipitation isotope seasonality on coastal western Greenland. *Journal of Geophysical Research: Atmospheres*, 126, e2020JD033543. <https://doi.org/10.1029/2020JD033543>
- Christiansen, B., & Ljungqvist, F. C. (2019). Challenges and perspectives for large-scale temperature reconstructions of the past two millennia. *Reviews of Geophysics* (1985), 55(1). <https://doi.org/10.1002/2016rg000521>
- Delhasse, A, Hanna, E, Kittel, C, Fettweis, X. Brief communication: CMIP6 does not suggest any atmospheric blocking increase in summer over Greenland by 2100. *Int J Climatol*. 2021; 41: 2589– 2596. <https://doi.org/10.1002/joc.6977>
- Fettweis, X., Franco, B., Tedesco, M., van Angelen, J. H., Lenaerts, J. T. M., van den Broeke, M. R., and Gallée, H.: Estimating the Greenland ice sheet surface mass balance contribution to future sea level rise using the regional atmospheric climate model MAR, *The Cryosphere*, 7, 469–489, <https://doi.org/10.5194/tc-7-469-2013>, 2013
- Furukawa, R., Uemura, R., Fujita, K., Sjolte, J., Yoshimura, K., Matoba, S., & Iizuka Y. (2017). Seasonal-scale dating of a shallow ice core from Greenland using oxygen isotope matching

between data and simulation. *Journal of Geophysical Research: Atmospheres*, 122. <https://doi.org/10.1002/2017JD026716>

- Graeter, K. A., Osterberg, E. C., Ferris, D. G., Hawley, R. L., Marshall, H. P., Lewis, G., et al. (2018). Ice core records of West Greenland melt and climate forcing. *Geophysical Research Letters*, 45, 3164– 3172. <https://doi.org/10.1002/2017GL076641>
- Hahn, L., Ummenhofer, C. C., & Kwon, Y.-O. (2018). North Atlantic natural variability modulates emergence of widespread Greenland melt in a warming climate. *Geophysical Research Letters*, 45, 9171– 9178. <https://doi.org/10.1029/2018GL079682>
- Hanna, E., Cropper, T.E., Hall, R.J. and Cappelen, J. (2016), Greenland Blocking Index 1851–2015: a regional climate change signal. *Int. J. Climatol.*, 36: 4847-4861. <https://doi.org/10.1002/joc.4673>
- Haris and Simons, 2012, <https://doi.org/10.1073/pnas.1206785109>
- Hawley, R., Courville, Z., Kehrl, L., Lutz, E., Osterberg, E., Overly, T., & Wong, G. (2014). Recent accumulation variability in northwest Greenland from ground-penetrating radar and shallow cores along the Greenland Inland Traverse. *Journal of Glaciology*, 60(220), 375-382. doi:10.3189/2014JoG13J141
- Holme, C., Gkinis, V., Lanzky, M., Morris, V., Olesen, M., Thayer, A., Vaughn, B. H., and Vinther, B. M.: Varying regional $\delta^{18}\text{O}$ -temperature relationship in high-resolution stable water isotopes from east Greenland, *Clim. Past*, 15, 893–912, <https://doi.org/10.5194/cp-15-893-2019>, 2019
- Johnsen, S. J., Dahl-Jensen, D., Gundestrup, N., Steffensen, J. P., Clausen, H. B., Miller, H., Masson-Delmotte, V., Sveinbjörnsdóttir, A. E. and White, J. 2001. Oxygen isotope and palaeotemperature records from six Greenland ice-core stations: Camp Century, Dye-3, GRIP, GISP2, Renland and NorthGRIPJ. *Quaternary Sci.*, Vol. 16, pp. 299–307. ISSN 0267-8179.
- Karlsson Nanna B., Eisen Olaf, Dahl-Jensen Dorthe, Freitag Johannes, Kipfstuhl Sepp, Lewis Cameron, Nielsen Lisbeth T., Paden John D., Winter Anna, Wilhelms Frank; Accumulation Rates during 1311–2011 CE in North-Central Greenland Derived from Air-Borne Radar

Data; *Frontiers in Earth Science*, Volume 4, 2016, <https://www.frontiersin.org/article/10.3389/feart.2016.00097>

- Keegan et al., 2014; <https://www.pnas.org/doi/full/10.1073/pnas.1405397111>
- Kjær, H. A., Zens, P., Edwards, R., Olesen, M., Mottram, R., Lewis, G., Terkelsen Holme, C., Black, S., Holst Lund, K., Schmidt, M., Dahl-Jensen, D., Vinther, B., Svensson, A., Karlsson, N., Box, J. E., Kipfstuhl, S., and Vallelonga, P.: Recent North Greenland temperature warming and accumulation, *The Cryosphere Discuss.* [preprint], <https://doi.org/10.5194/tc-2020-337>, 2021.
- Kunz, T., & Laepple, T. (2021). Frequency-Dependent Estimation of Effective Spatial Degrees of Freedom. *Journal of Climate*, 34(18), 7373–7388. <https://doi.org/10.1175/JCLI-D-20-0228.1>
- Lenaerts, J. T. M., Le Bars, D., van Kampenhout, L., Vizcaino, M., Enderlin, E. M., and van den Broeke, M. R. (2015), Representing Greenland ice sheet freshwater fluxes in climate models, *Geophys. Res. Lett.*, 42, 6373–6381, doi:10.1002/2015GL064738.
- Ljungqvist, F. C., Krusic, P. J., Brattström, G., and Sundqvist, H. S.: Northern Hemisphere temperature patterns in the last 12 centuries, *Clim. Past*, 8, 227–249, <https://doi.org/10.5194/cp-8-227-2012>, 2012. McKay, N. P. & Kaufman, D. S. An extended Arctic proxy temperature database for the past 2,000 years. *Scientific Data* 1, 140026, doi:10.1038/sdata.2014.26 (2014).
- McLeod, Jordan T. and Thomas L. Mote. “Linking interannual variability in extreme Greenland blocking episodes to the recent increase in summer melting across the Greenland ice sheet.” *International Journal of Climatology* 36 (2016): n. Pag.
- Neukom, R., Steiger, N., Gómez-Navarro, J. J., Wang, J., & Werner, J. P. (2019). No evidence for globally coherent warm and cold periods over the preindustrial Common Era. *Nature*, 571(7766), 550. <https://doi.org/10.1038/s41586-019-1401-2>
- Nghiem, S. V., Hall, D. K., Mote, T. L., Tedesco, M., Albert, M. R., Keegan, K., Shuman, C. A., DiGirolamo, N. E., and Neumann, G. (2012), The extreme melt across the Greenland ice sheet in 2012, *Geophys. Res. Lett.*, 39, L20502, doi:[10.1029/2012GL053611](https://doi.org/10.1029/2012GL053611).

- Nicholas A. Kamenos, Trevor B. Hoey, Peter Nienow, Anthony E. Fallick, Thomas Claverie; Reconstructing Greenland ice sheet runoff using coralline algae. *Geology* 2012;; 40 (12): 1095–1098. doi: <https://doi.org/10.1130/G33405.1>
- Orsi, A., Kawamura, K., Fegyveresi, J., Headly, M., Alley, R., & Severinghaus, J. (2015). Differentiating bubble-free layers from melt layers in ice cores using noble gases. *Journal of Glaciology*, 61(227), 585-594. doi:10.3189/2015JoG14J237
- Overland, J., Francis, J. A., Hall, R., Hanna, E., Kim, S., & Vihma, T. (2015). The Melting Arctic and Midlatitude Weather Patterns: Are They Connected?, *Journal of Climate*, 28(20), 7917-7932. Retrieved Jun 26, 2022, from <https://journals.ametsoc.org/view/journals/clim/28/20/jcli-d-14-00822.1.xml> PAGES 2k Consortium. Continental-scale temperature variability during the past two millennia. *Nature Geosci* 6, 339–346 (2013). <https://doi.org/10.1038/ngeo1797>
- Taranczewski, T., Freitag, J., Eisen, O., Vinther, B., Wahl, S., and Kipfstuhl, S.: 10,000 years of melt history of the 2015 Renland ice core, EastGreenland, *The Cryosphere Discuss.* [preprint], <https://doi.org/10.5194/tc-2018-280>, 2019.
- Vallelonga, P., Christianson, K., Alley, R. B., Anandakrishnan, S., Christian, J. E. M., Dahl-Jensen, D., Gkinis, V., Holme, C., Jacobel, R. W., Karlsson, N. B., Keisling, B. A., Kipfstuhl, S., Kjær, H. A., Kristensen, M. E. L., Muto, A., Peters, L. E., Popp, T., Riverman, K. L., Svensson, A. M., Tibuleac, C., Vinther, B. M., Weng, Y., and Winstrup, M.: Initial results from geophysical surveys and shallow coring of the Northeast Greenland Ice Stream (NEGIS), *The Cryosphere*, 8, 1275–1287, <https://doi.org/10.5194/tc-8-1275-2014>, 2014.
- Vinther, B. M., Andersen, K. K., Jones, P. D., Briffa, A. K. R., & Cappelen, J. (2006). Extending Greenland temperature records into the late eighteenth century. *Journal of Geophysical Research-atmospheres*, 111(D11), [D11105]. <https://doi.org/10.1029/2005JD006810>
- Vinther, B. M., et al. (2006), A synchronized dating of three Greenland ice cores throughout the Holocene, *J. Geophys. Res.*, 111, D13102, doi:[10.1029/2005JD006921](https://doi.org/10.1029/2005JD006921).
- Vinther, 2011; PAGES news • Vol 19 • No 1 • March 2011
- B.M. Vinther, P.D. Jones, K.R. Briffa, H.B. Clausen, K.K. Andersen, D. Dahl-Jensen, S.J. Johnsen, Climatic signals in multiple highly resolved stable isotope records from

Greenland, *Quaternary Science Reviews*, Volume 29, Issues 3–4, 2010, Pages 522-538, ISSN 0277-3791, <https://doi.org/10.1016/j.quascirev.2009.11.002>.

- Westhoff, J., Sinnl, G., Svensson, A., Freitag, J., Kjær, H. A., Vallelonga, P., Vinther, B., Kipfstuhl, S., Dahl-Jensen, D., and Weikusat, I.: Melt in the Greenland EastGRIP ice core reveals Holocene warm events, *Clim. Past*, 18, 1011–1034, <https://doi.org/10.5194/cp-18-1011-2022>, 2022.
- Zuhr, A. M., Münch, T., Steen-Larsen, H. C., Hörhold, M., and Laepple, T.: Local-scale deposition of surface snow on the Greenland ice sheet, *The Cryosphere*, 15, 4873–4900, <https://doi.org/10.5194/tc-15-4873-2021>, 2021.

Reviewer Reports on the First Revision:

Referees' comments:

Referee #1 (Remarks to the Author):

The authors provided very comprehensive and satisfactory responses to my initial review, and have significantly expanded and improved several sections of the manuscript in great detail. Adding the newer version of MAR and extending the period of analysis for meltwater vs temperature to 1871 adds further temporal perspective on just how unusual early 21st century warmth and melt has been. Their robust statistical approach clearly distinguishes 21st century warming as exceptional with respect to the last millennium and establishes that there was virtually no possibility of such warmth and melt to occur prior to anthropogenic warming. The authors are careful to disentangle how broad, spatially, their dataset can represent temperatures across the Arctic — much of what context they can provide is quite Greenland specific due to the unique nature of that location (elevation, albedo, temperature, etc.) with respect to the broader Arctic land surface. This revised manuscript greatly improves on the prior iteration, and is an excellent example of how ice core arrays provide critical reconstructions of past climate variability that inform future projections — in this case, that continued warming will bring increased meltwater runoff that continues to be exceptional with respect to the last 1000 years.

Referee #2 (Remarks to the Author):

Second Review of Horhold et al., Exceptional temperatures in central-north Greenland

Horhold et al. have provided a revised manuscript with detailed responses. The primary improvement to the manuscript is establishing a quantitative relationship with the meltwater runoff. This considerably strengthens the paper and increases the utility of the NGT-2012.

Overall, the authors have addressed my concerns. The statistical treatments are robust. I was glad to see the accumulation rate reconstructions added; even though the accumulation did not provide a clear enough interpretation to be shown in the main paper, they will be a useful record for other researchers.

I have just two points to raise:

1) Figure 4 has been significantly improved overall. I am wondering if the NGT2012 pre-industrial is capturing the variability accurately because I am surprised that so many years from 1880 exceed the $p=0.95$ contour – and not because of abnormally warm temperatures. Is averaging of water isotopes without perfect timescales artificially limiting the variability in NGT2012? Or is something else happening with melt water runoff? i.e. Greenland Blocking has a big influence on melt water runoff even without warmer temperatures?

2) The authors provided a strong response to my first comment about why a north-central Greenland temperature record is of outsized importance relative to its geographic area. The point about not needing to calibrate to instrumental records is an important one. I think this is a point

worth making more strongly in the opening of the paper.

Minor points:

L33-35 – This is a very general statement that I think can be made stronger, as discussed in point 2 above.

L43 – delete “only”

L45 – is the paleoclimate data from Greenland really sparse compared to other regions? It seems like Greenland is actually very well covered with high resolution records

F.g 1C – anomaly to label “Melt Runoff Anomaly”

L208 – reword “the major single source”. I think you mean that Greenland runoff now exceeds thermal expansion and runoff from all other glaciers and ice sheets. If so, I think you should write that out.

L226 – delete “This shows that” to tighten the writing.

L320 – A discussion of why the LGM-Holocene value of 0.33 is not considered in the isotope-temperature calibration? I can see reasons to exclude this possibility, but it’s a prominent enough result that its worth discussing the reasoning for why it wouldn’t apply to the past 1 ka.

Reviewer Reports on the First Revision:

Answer to Reviewer 2

I have just two points to raise:

- 1) Figure 4 has been significantly improved overall. I am wondering if the NGT2012 pre-industrial is capturing the variability accurately because I am surprised that so many years from 1880 exceed the $p=0.95$ contour – and not because of abnormally warm temperatures. Is averaging of water isotopes without perfect timescales artificially limiting the variability in NGT2012? Or is something else happening with melt water runoff? i.e. Greenland Blocking has a big influence on melt water runoff even without warmer temperatures?

Answer:

We thank the reviewer for pointing to this observation. As we detail below, this behavior is not caused by an underestimation of the past NGT2012 variability but influenced by an early (after 1800) increase of the NGT2012 isotope values compared to the reference period (1000-1800CE).

In principle, multiple processes could lead to an underestimation of past variability. These are (1) a changing number of records, (2) more diffusion in the older and thus deeper part of the cores or (3) chronological uncertainty. As this could affect our conclusions, already in the early stage of the analyses, we carefully checked each of the processes We used ‘frozen stacks’ and artificially diffused all cores and described the results in the manuscript (Extended Figure 3 and Method sections). The time-uncertainty is damping the variability on interannual scales but does not affect the decadal scales analyzed here; this is also visible in the high SNR estimate at decadal scales (Extended Data Figure 1b). Further, the uncertainty is always minimal at the volcanic reference horizons that are roughly equally distributed through time and thus does not accumulate further back in time (Table 2 of Weissbach et al., Climate of the Past 2016 provides the used horizons). Analyzing the change in variance of NGT2012 through time shows changes over time but no systematic decrease back in time (Fig. R1). Thus, there is no evidence of a variance loss further back in time.

Figure R1: Variance estimated on a running 50yr window on the NGT2012 stack; While the estimated variance varies over time (as also expected from estimating a variance of a running window on a stationary stochastic process) there is no evidence for a systematic decrease in the earlier time periods.

The $p = 0.95$ and $p = 0.99$ contour lines in Figure 4b are derived from the pre-industrial distribution (1000-1800). In contrast, the colored points show the MAR3.5.2 meltwater runoff anomalies of the 1871-2011 CE period vs the respective NGT-2012 temperature anomalies i.e. a time interval not included in the probability estimates. NGT2012 shows an early (after 1800) increase in d18O (Figure 1a), which might be potentially already related to anthropogenic forcing (Abram et al., 2016) although it is not statistically significant in our dataset (mean anomaly 1000-1800; -0.17; 1800-1950: -0.1, $p > 0.1$). This leads to more outliers than expected from the pre-industrial distribution including the warm events from 1880 and 1930 (see also Figure 1 c).

The choice of the pre-industrial period until 1800 follows earlier studies, as there is no objective starting date of the industrial area. We have tested the robustness of the found probability for the 2001-2011 value to occur using different pre-industrial time periods, see Methods, Extended Data Table 2, and find the probability still to be extremely small, when defining pre-industrial until 1900.

As the reviewer points out increased values for meltwater runoff can be related to other influences than temperature only such as Greenland blocking (Figure ED 2d and Figure ED 2f) and this is likely the reason why some enhanced runoff values in the historical record are not related to enhanced temperatures. However, these effects (when stationary over time) are already included in the uncertainty term of our meltwater run-off reconstruction and thus the size of the ellipses. Finally, in response to the reviewer, we tested again our methodology (the empirical coverage of the quantiles) using stationary surrogate data with the same properties (autocorrelation and correlation structure) as the true data. This confirms that we get the right coverage in expectation (Figure R2), however in single realizations, there can be more or less outliers than expected on average.

Figure R2: Reconstructed meltwater runoff anomalies over temperature anomalies from surrogate data with similar autocorrelation structure and correlation as NGT-2012 and MAR3.5.2 meltwater runoff anomalies.

In summary, the observation that multiple years from 1880 CE on exceed the $p=0.95$ and $p=0.95$ contour likely results from the already increasing temperatures since 1800 and are partly by chance. This does not affect our finding that our NGT-2012 temperature record from central Greenland significantly correlates to Greenland meltwater run-off and can be used to derive the presented reconstruction for pre-industrial times.

Changes in the manuscript:

We added a note to the caption of Figure 4: “ We note the high number of outliers is likely due to the increase of NGT-2012 after 1800 (Figure 1) that might be related to the early onset of industrial-era warming (Abrams et al., 2016).”

We did not add details on the variance preservation, non-temperature effects on run-off and method evaluation in the manuscript as we feel that the former is already included in the manuscript (Figure ED 3, Figure ED 2d and Figure ED 2f and Method sections) and the latter was just a ‘sanity check’ to show that the method works as expected.

References:

- Weißbach, S., Wegner, A., Opel, T., Oerter, H., Vinther, B. M., and Kipfstuhl, S.: Spatial and temporal oxygen isotope variability in northern Greenland – implications for a new climate record over the past millennium, *Clim. Past*, 12, 171–188, <https://doi.org/10.5194/cp-12-171-2016>, 2016.
- Abram, N. J. *et al.* Early onset of industrial-era warming across the oceans and continents. *Nature* **536**, 411-418, doi:10.1038/nature19082 (2016).

2) The authors provided a strong response to my first comment about why a north-central Greenland temperature record is of outsized importance relative to its geographic area. The point about not needing to calibrate to instrumental records is an important one. I think this is a point worth making more strongly in the opening of the paper.

Answer:

We add a related sentence in the Introduction.

Changes in the manuscript:

In Line 34ff we added: *“However, most large-scale reconstructions based on a multiple proxy types or tree ring records require a proxy screening and instrumental calibration step and thus might be prone to underestimation of past climate variability outside of the calibration period.”*

Minor points:

L33-35 – This is a very general statement that I think can be made stronger, as discussed in point 2 above.

Answer: We added a sentence here, see comment above.

L43 – delete “only”

We deleted “only”

L45 – is the paleoclimate data from Greenland really sparse compared to other regions? It seems like Greenland is actually very well covered with high resolution records

Answer:

We need spatio-temporal coverage, as single ice cores contain too much noise to obtain natural variability and a representative climate signal. Spatial arrays to display regional climate are sparse. Central- and North Greenland are only covered by NGT, presented here.

F.g 1C – anomaly to label “Melt Runoff Anomaly”

Answer:

In fact, all the time series in Figure 1 display anomalies relative to 1961-90, but none of the legends states this explicitly in order to keep them concise. In this respect the figure labels are consistent. In addition, the figure legend currently mentions for every individual data set that we use anomaly time series. However, in order to state this more prominently, we revised the legend by adding an extra sentence at the end. In addition, we added horizontal dashed lines to Figure 1c, similar to panel a, marking the reference value.

Changes in the manuscript:

- We added a sentence at the end of the figures legend: *“All time series are displayed as anomalies relative to the 1961-1990 CE reference period (horizontal dashed lines).”*
- We added horizontal dashed line to Figure 1c, similar to Figure 1a, marking the reference value.

L208 – reword “the major single source”. I think you mean that Greenland runoff now exceeds thermal expansion and runoff from all other glaciers and ice sheets. If so, I think you should write that out.

Changes in the manuscript:

We have rephrased the sentence to: *“Greenland has become a major source of mass-related sea level rise in the past decade, exceeding thermal expansion and contribution from other glaciers, due to...”*

L226 – delete “This shows that” to tighten the writing.

We deleted “This shows that”. It reads now: *“(…) allows us to generate the first reconstruction of the meltwater runoff anomalies for Greenland over the last millennium and thus to put the recent runoff anomalies in the long-term context (Figure 4b, Methods). The meltwater runoff anomalies of the 2001 – 2011 decade...()”*

L320 – A discussion of why the LGM-Holocene value of 0.33 is not considered in the isotope-temperature calibration? I can see reasons to exclude this possibility, but it’s a prominent enough result that its worth discussing the reasoning for why it wouldn’t apply to the past 1 ka.

It has been shown, that the use of present-day observations (spatial slope) do not represent glacial – interglacial surface temperature changes in Central Greenland (Jouzel et al., 2003, Krinner and Werner, 2003), i.e. the temperature-isotope relationship differs between LGM and Holocene. As reasons a change in seasonality of the precipitation is discussed (e.g. Jouzel et al.,1997, 2000) as well as changes in the moisture source region (Jouzel et al.,1997) .

Therefore the LGM – Holocene slope is not used for present-day reconstructions.

Changes in the manuscript:

Method, Temperature calibration of the NGT-2012 stack: We added a sentence in the method section: *“We do not apply any LGM-Holocene temporal slope, as it is not representative for present-day conditions (Krinner and Werner, 2003) due to different seasonality in precipitation or moisture source during the LGM (Jouzel et a., 1997, 2000).*

References

- G. Krinner, M. Werner, Impact of precipitation seasonality changes on isotopic signals in polar ice cores: a multi-model analysis, *Earth and Planetary Science Letters*, Volume 216, Issue 4, 2003, Pages 525-538, ISSN 0012-821X, [https://doi.org/10.1016/S0012-821X\(03\)00550-8](https://doi.org/10.1016/S0012-821X(03)00550-8).
- Jouzel, J., et al. (1997), Validity of the temperature reconstruction from water isotopes in ice cores, *J. Geophys. Res.*, 102(C12), 26471– 26487, doi:[10.1029/97JC01283](https://doi.org/10.1029/97JC01283).
- J. Jouzel, G. Hoffmann, R.D. Koster, V. Masson, Water isotopes in precipitation:: data/model comparison for present-day and past climates, *Quaternary Science Reviews*, Volume 19, Issues 1– 5, 2000, Pages 363-379, ISSN 0277-3791, [https://doi.org/10.1016/S0277-3791\(99\)00069-4](https://doi.org/10.1016/S0277-3791(99)00069-4). (<https://www.sciencedirect.com/science/article/pii/S0277379199000694>)